# A mechanical-coupling mechanism in OSCA/TMEM63 channel mechanosensitivity

Mingfeng Zhang [1,2,5] ✉, Yuanyue Shan[1,2,5], Charles D. Cox [3,4] ✉ & Duanqing Pei [2] ✉

Mechanosensitive (MS) ion channels are a ubiquitous type of molecular force sensor sensing forces from the surrounding bilayer. The profound structural diversity in these channels suggests that the molecular mechanisms of force sensing follow unique structural blueprints. Here we determine the structures of plant and mammalian OSCA/TMEM63 proteins, allowing us to identify essential elements for mechanotransduction and propose roles for putative bound lipids in OSCA/TMEM63 mechanosensation. Briefly, the central cavity created by the dimer interface couples each subunit and modulates dimeric OSCA/TMEM63 channel mechanosensitivity through the modulating lipids while the cytosolic side of the pore is gated by a plug lipid that prevents the ion permeation. Our results suggest that the gating mechanism of OSCA/TMEM63 channels may combine structural aspects of the 'lipid-gated' mechanism of MscS and TRAAK channels and the calcium-induced gating mechanism of the TMEM16 family, which may provide insights into the structural rearrangements of TMEM16/TMC superfamilies.

Force sensitive proteins can respond to mechanical force which breaks a conformational energy barrier enabling them to convert force into a cellular signal[1]. One of the principal types of force sensitive proteins are mechanosensitive (MS) ion channels[2,3]. MS channels sense mechanical force which they subsequently convert into an electrical signal[3]. They are widely distributed among the prokaryotic and eukaryotic kingdoms and are central to the molecular basis of many mechanosensory processes such as touch, and hearing[4,5]. Most MS channels share the ability to transduce forces from the surrounding bilayer[6–9], a concept known as force-from-lipids[10]. This is despite the fact that the architectures of these families of ion channels are strikingly different[3,11]. Thus, it is likely that MS channels have developed unique molecular mechanisms to sense mechanical force.

OSCA/TMEM63 channels[12,13] represent a more recently identified class of inherently mechanosensitive channels. This family of channels is conserved from yeast to humans, playing important roles in plant osmo-sensing[14], insect food texture sensing[15], mouse hearing[16], and

brain function[17]. OSCA/TMEM63 channels such as *AtOSCA1.1*, *AtOSCA1.2* and *AtOSCA3.1* from *Arabidopsis thaliana* assemble as dimers[13,18]. Their architectures share structural similarity with the TMEM16 superfamily, including the calcium activated anion channel and calcium activated scramblases[13]. In TMEM16A channels, calcium ions bind to the cytosolic segment of the pore-lining M6 helix which triggers anion flow across the cell membrane[19]. The OSCA/TMEM63 channels also share structural similarity with TMC proteins, which play a pivotal role in the auditory system[20,21] and may also act as lipid scramblases[22]. Therefore, studying the gating cycle of these channels not only helps us to dissect the molecular mechanism(s) of OSCA/TMEM63 channel gating and function but may also provide important clues to understand other structurally similar proteins.

In this study, we show that the mechanosensitive mammalian homologs of OSCA channels the TMEM63A/B channels are monomeric channels when purified and structurally characterized in a detergent environment, suggesting that one subunit of the OSCA/TMEM63

[1]Fudan University, Shanghai 200433, China. [2]Laboratory of Cell Fate Control, School of Life Sciences, Westlake University, Hangzhou 310000, China. [3]Victor Chang Cardiac Research Institute, Sydney 2010, Australia. [4]School of Biomedical Sciences, Faculty of Medicine & Health, UNSW Sydney, Kensington, New South Wales 2052, Australia. [5]These authors contributed equally: Mingfeng Zhang, Yuanyue Shan. ✉e-mail: zhangmingfeng@westlake.edu.cn; c.cox@victorchang.edu.au; peiduanqing@westlake.edu.cn

channel core structure is likely sufficient to form an MS channel. The cytosolic side of the pore of TMEM63A is plugged by a lipid-like density which would need to move to allow ion permeation. To see whether similar lipid-like densities were present in the pore region of OSCA channels we solved the structure of the *Arabidopsis thaliana* OSCA1.1 (*At*OSCA1.1) channel in a lipidic environment. In doing so not only did we identify similar lipid-like densities under the pore we also observed that the central cavity was extended by the insertion of two putative lyso-lipid molecules, resulting in the motion of M0 and M6 providing a pathway to channel activation. We identified similar extended and contracted states in the less pressure sensitive homolog *At*OSCA3.1. Furthermore, the dimer interface of the *At*OSCA3.1 in the contracted state was 'locked' by the interaction between each single subunit through the M5 and M6 linker, presumably providing an additional energy barrier to overcome during activation. These results indicate that the central cavity created by the dimer interface couples each subunit and modulates OSCA channel mechanosensitivity. Combined our data suggests that the gating mechanism of OSCA/TMEM63 channels may combine structural aspects of the 'lipid-gated' mechanism of MscS and TRAAK channels and the calcium-induced gating mechanism of the TMEM16 family. Our structural characterization of OSCA/TMEM63 channels provides a detailed framework to understand OSCA/TMEM63 channel mechanosensitivity and may provide insights into the structural rearrangements seen in the related TMEM16/TMC superfamilies.

## Results

### Mammalian OSCA channels are monomers

To explore the potential force sensing mechanism of mechanosensitive OSCA/TMEM63 channels which are conserved from yeast to humans, we solved OSCA/TMEM63 channel structures from different species as they have different channel properties including pressure threshold and channel conductance[12,13]. Two important mammalian OSCA/TMEM63 homologs are *h*TMEM63A and *m*TMEM63B, which contribute to hearing and brain function, respectively[16,17]. We expressed the human and mouse orthologues of *h*TMEM63A and *m*TMEM63B with a C-terminal GFP tag in HEK293F cells and extracted the proteins from membranes using a combination of lauryl maltose neopentyl glycol (LMNG) and cholesteryl hemisuccinate (CHS)[13]. Surprisingly, we found, regardless of the detergent used for extraction (LMNG, DDM: n-Dodecyl β-D-maltoside, GDN: Glyco-diosgenin), that *h*TMEM63A and *m*TMEM63B were monomeric. This became especially clear when comparing them with the dimeric *At*OSCA1.1 using fluorescence-based size-exclusion chromatography (FSEC) (Supplementary Fig. 1a, b). After standard protein preparation, cryo-EM sample preparation, image collection, and data processing, we solved the cryo-EM structure of *h*TMEM63A and *m*TMEM63B in a detergent environment at an overall resolution of 4.0 Å for *m*TMEM63B (Supplementary Fig. 2a–f and Supplementary Table 1) and 3.3 Å for *h*TMEM63A (Supplementary Fig. 3a–f and Supplementary Table 1). The near atomic resolution map of *h*TMEM63A allowed us to build the respective monomeric structural models (Fig. 1a, b, Supplementary Figs. 9d and 10a, b). The organization of *m*TMEM63B is almost identical to the transmembrane architecture of *h*TMEM63A (Supplementary Fig. 4a–c), indicative of the conserved structure and function in mammals. Therefore, given the higher resolution of the *h*TMEM63A structure we restricted our below discussion almost entirely to *h*TMEM63A. Structural comparison of the monomeric *h*TMEM63A and *At*OSCA1.1, demonstrates that the M10 of *h*TMEM63A is rotated into the 'central cavity' and would sterically hinder the dimeric interface (Fig. 1c, d), providing a likely reason for the monomeric form. The electrostatic and hydrogen bonding networks that occur in *At*OSCAs specifically those at the cytosolic dimeric interface are not conserved in *h*TMEM63A or *m*TMEM63B (Fig. 1e), also consistent with the monomeric structure of mammalian OSCA/TMEM63 channels.

### A "lipid plug" occludes the pore in *h*TMEM63A

The monomeric *h*TMEM63A channel shares structural similarity with the single subunit of *At*OSCA1.1 channels (Fig. 1c, d). The two anti-parallel helices of the two-blade propeller at the intracellular surface shape sits at the level of the membrane interface and likely forms protein-lipid interactions (Fig. 1a, b). Like *At*OSCA1.1[13], the putative narrow pore of *h*TMEM63A is formed by the M3, M4, M5, M6, and M7 helices, which is a typical hourglass-like pore (Fig. 2a–d). The tightest point along the permeation pathway in the *h*TMEM63A pore occurs around the conserved hydrophobic residues F555, F556, and Y559 on M6 (Fig. 2d, e), with the narrowest section of the pore being less than 2 Å (Fig. 2c, d). The relevant contributing residues from M3, M4, and M5 are also shown for completeness (Fig. 2d, e). Given the hydrophobicity of this section the *h*TMEM63A structure likely represents a non-conducting state. The narrow pore is consistent with the small unitary conductance of TMEM63 channels[12]. Unlike the *At*OSCA1.1 channel, the *h*TMEM63A pore region at the extracellular side is shielded by an extra N-terminal soluble domain that extends from M0 and is stabilized by the M3 and M4 linker and the M1 and M2 linker (Supplementary Fig. 4d–f). This may also contribute to the significantly smaller conductance of *h*TMEM63A when compared to *At*OSCA1.1[12]. Interestingly, we found a lipid-like density that plugged the cytosolic entrance of the pore at the intersection of M4 and M6 (Fig. 2a–c). The coordinating residues of *h*TMEM63A that interacted with the putative phosphate of the lipid were W473 on M4, E571 on M6, and W613 on M7, while hydrophobic residues on M0 and M4 interact with the acyl chain (Fig. 2f). The corresponding residues that would interact with the "lipid plug" in *At*OSCA1.1 are the polar amino acids K436 and Q573 instead of the hydrophobic amino acid of W473 on M4 and W613 on M7, respectively (Supplementary Fig. 4g–i). The site of this "lipid plug" is equivalent to the calcium-binding sites identified in the TMEM16 family[23,24] (Supplementary Fig. 4j, k). It suggests that this putative lipid may play a vital role in *h*TMEM63 channel gating and resembles the lipids that occlude the pore in both the MscS[25–27] and TRAAK[8].

### An extended state of *AtOSCA*1.1 in lipid nanodiscs

As lipids play a critical role in MS channel gating[25,28–31] and to see if we could identify lipids such as the lipid plug in TMEM63 we next embedded the purified *AtOSCA*1.1 channel protein in a lipid nanodisc environment. Then we solved the *AtOSCA*1.1 channel structure in the lipid nanodisc environment at 2.5 Å resolution. The 2.5 Å resolution map allowed us to build the model of nearly full-length *AtOSCA*1.1 (Supplementary Figs. 5a–g, 9a and Supplementary Table 1). The *AtOSCA*1.1 channel in a lipid nanodisc is in an overall extended state when compared to the detergent environment (PDB:6JPF; Fig. 3a–e). We used the parallelogram of the midpoint of the transmembrane segments of *AtOSCA*1.1 to represent the central cavity area, specifically the Cα atoms of the F384 and I660 residues (Fig. 3f). This illustrated that the area of the central cavity was extended (Fig. 3f). The extended motion resembles the prestin protein which senses voltage and underlies electromotility in the auditory system which undergoes and even more pronounced movement[32,33]. In addition, the structurally similar TMC proteins which play a central role in auditory transduction also exhibit an extended motion at the dimer interface[34].

We found a strong lipid density on the extracellular side of the central cavity and a lyso-lipid-like molecule fitted this density well (Fig. 3b). The putative lyso-lipids seem to insert into the central cavity and extend the central cavity like a rivet docking in the huge lipid storage cavity, thus resulting in the extension of the lipid storage cavity and allosterically regulating M6 resulting in channel activation (Fig. 3c), which is consistent with our previous electrophysiological finding that lyso-lipids promote OSCA channel activation[13]. As the putative lyso-lipid in the central cavity may regulate the channel activity, we refer to it as a 'modulating' lipid. We thus mutated coordinating residues of this bound lipid and interrogated their

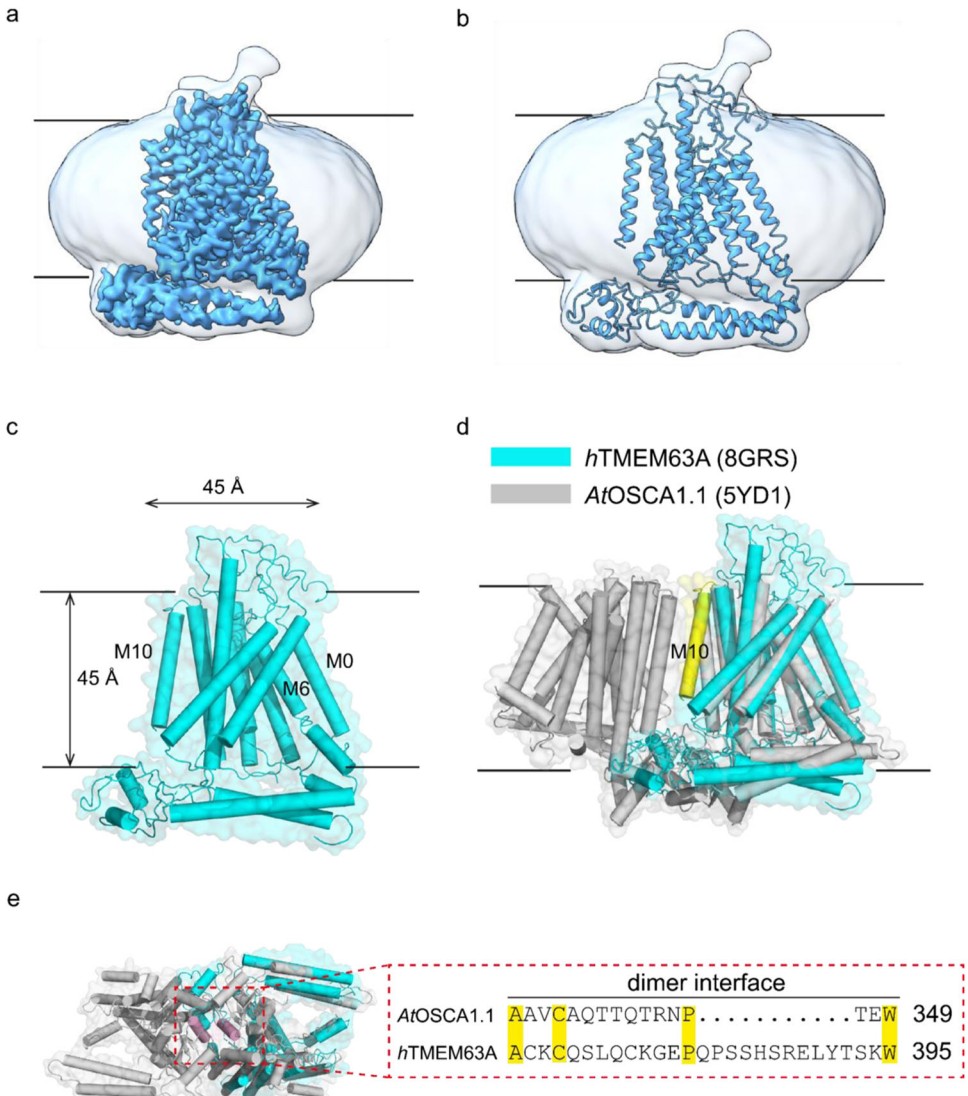

**Fig. 1 | Overall structure of the monomeric hTMEM63A channel. a–c** Cryo-EM density map (**a**), cartoon representation (**b**), and surface representation (**c**) showing the side view of the hTMEM63A channel. The approximate extent of the phospholipid bilayer is shown as thin black lines. The detergent micelle is shown in white with a low pass filter and hTMEM63A channel is shown in blue in (**a**, **b**). Helices are shown as cylinders and hTMEM63A channel is shown in cyan in (**c**). **d** Structure comparison of hTMEM63A and AtOSCA1.1 (PDB: 6JPF) is shown as a cartoon representation. AtOSCA1.1 is shown in gray. The monomeric hTMEM63A channel (shown in cyan) shares structural similarity to the single subunit of AtOSCA1.1 with the M10 helix highlighted in yellow. **e** Comparison of the sequence differences at the dimer interface of AtOSCA1.1 (gray) and hTMEM63A (cyan). The region shown in the alignment is colored purple for clarity.

mechanosensitivity. The mutation of H651A (a putative lyso-lipid coordinating residue) right shifts the pressure response curve indicative of a channel less sensitive to mechanical force (Fig. 3g–j).

### The contracted and extended states of AtOSCA3.1

Inspired by the observation of the extended state of AtOSCA1.1 channels in the nanodisc, we attempted to solve the high-resolution structure of another member of the OSCA/TMEM63 channel family, the higher pressure activated mechanosensitive atOSCA3.1 channel which exhibits a half activation pressure threshold that is much higher than that of AtOSCA1.1. Our high-resolution cryo-EM data obtained in a detergent environment contain two conformations of AtOSCA3.1 channels (Supplementary Figs. 6a–g, 7a, b, 9b, c and Supplementary Table 1). One is identical to our previously low-resolution model of AtOSCA3.1[13], which we call an extended state. The other is a more compact contracted state (Fig. 4a–f). We analyzed each monomer of AtOSCA3.1 in the two states and found that the single monomer of the contracted state superimposes with the extended state

(Supplementary Fig. 7c). Therefore, the major conformational change occurs in the central cavity. In the contracted state, each subunit of the M5 and M6 linker forms another dimer interface through an L487 interaction (Fig. 4h, i). Therefore, the area of the central cavity is reduced compared to the extended state (Fig. 4g). The disruption of the M5 and M6 linker interaction and extension of the central cavity is likely an energy barrier for channel transition to the extended state and then opening of the channel.

## Discussion

### Multiple putative lipids modulate OSCA/TMEM63 channel gating

Based on electrophysiological studies, the "force-from-lipid" principle[10] is conserved in OSCA/TMEM63 channel gating[12,13]. In bacterial MscS, there are multiple lipids that reside in the channel pore[25–27]. Pulling theses lipids out of the pore, by an as yet unknown route, in response to increasing membrane tension is essential to channel opening[25]. A similar mechanism of lipids sealing the pore was originally

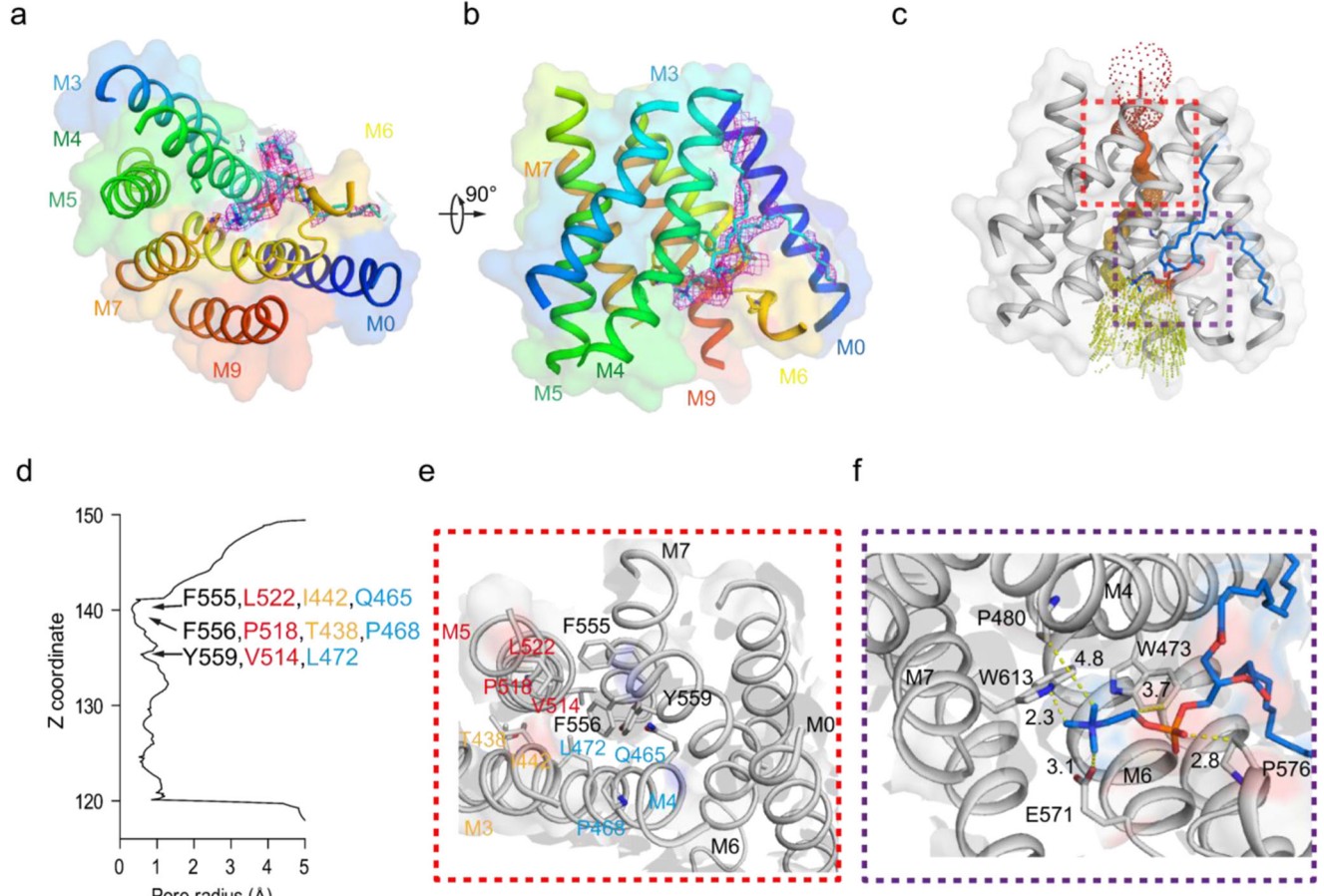

**Fig. 2 | The putative narrow pore and lipid binding site of the monomeric *h*TMEM63A channel. a**, **b** Top view (**a**) and side view (**b**) of the transmembrane domain of *h*TMEM63A channel. The transmembrane helices are marked by 0-9 and colored in the rainbow. The putative narrow pore of *h*TMEM63A is formed by the M3, M4, M5, M6, and M7. A lipid that plugs the cytosolic entrance of the pore (modeled as POPC) at the intersection of M4 and M6 is shown with its corresponding map density (σ = 3.0) in (**a**). **c** Side view of the *h*TMEM63A transmembrane domain. The calculated pore profile of the putative *h*TMEM63A is indicated as orange and yellow dots calculated using HOLE. The transmembrane helices in the pore domain are shown in gray. **d** Pore radius of the *h*TMEM63A along the z-axis. The positions of the hydrophobic residues F516, F517, and Y520 on M7 that

are blocking the pore are marked by arrows and the relative contributions to the pore of other pore-lining helices are shown with the same color coding as in panel (**e**). **e** Enlarged cutaway view of the narrow pore of *h*TMEM63A corresponding to the red box in (**c**). The transmembrane helices are in gray and the hydrophobic residues are labeled. **f** Enlarged cutaway view of lipid binding interface corresponding to the purple box in (**c**). The lipid is shown in blue and plugs the cytosolic entrance of the pore at the intersection of M4 and M6. Residues W473 on M4, E571 on M6, and W613 on M7 coordinately interact with the phosphate head of the putative lipid, while hydrophobic residues on M0 and M4 interact with the acyl chain (relative distances are shown in Å).

suggested for the human mechanosensitive TRAAK channel[8]. In the monomeric *h*TMEM63A channel, we identified a putative analogous pore lipid that plugs the cytosolic side of the permeation pathway drawing clear parallels with MscS and TRAAK. In addition, this "lipid plug" is in the corresponding region of the calcium-binding site of the TMEM16 family and interacts with M6 and M0 of the *h*TMEM63A channel suggesting that the gating mechanism may also share structural similarities with the activation mechanism of TMEM16 family. Moreover, not only did we observe the putative "plug lipid" in *h*TMEM63a, but we also observed lipid-like densities in a similar position in dimeric *At*OSCA1.1 and *At*OSCA3.1 channels, albeit not as structurally well resolved as that identified in *h*TMEM63a (Supplementary Fig. 8a–d). This raises the possibility that this putative lipid may play a similar role in the gating cycles of OSCA and TMEM63 channels.

In other MS channels such as bacterial MscS multiple structural lipids play roles in the gating cycle including pore lipids and other integral lipids such as the "gate-keeper" or "hook lipids"[25,26]. We also identified a second lipid-like density in the upper portion of the central cavity of *At*OSCA1.1. This seemed to correspond to a lyso-lipid. We

know from previous work that lyso-lipids can activate OSCA channels[13]. In TMEM16A, a second calcium-binding site is located within the dimer interface, which regulates channel activity through a long-range allosteric gating mechanism[35–37]. The addition of calcium at the dimeric interface of TMEM16A is reminiscent of the putative lyso-lipid in the central cavity of the dimeric *At*OSCA1.1 channel. Therefore, at least two key lipids may modulate the channel sharing a globally convergent mechanism with the TMEM16 superfamily. While the putative "plug-lipid" may be conserved within the OSCA/TMEM63 superfamily, the relevance of the lyso-lipid binding site is less clear for the monomeric TMEM63 channels. Future work should aim to characterize whether TMEM63 family members respond to conical lipids like lyso-phosphatidylcholine.

### Central cavity extension couples M0 and M6 motion during OSCA channel activation

The major conformational change leading to gating is likely the motion of M0 and M6 for *h*TMEM63A, *At*OSCA3.1, and *At*OSCA1.1 channels[13]. This is supported by our previous electrophysiological and structural results suggesting that the M0 and M6 were important channel gating

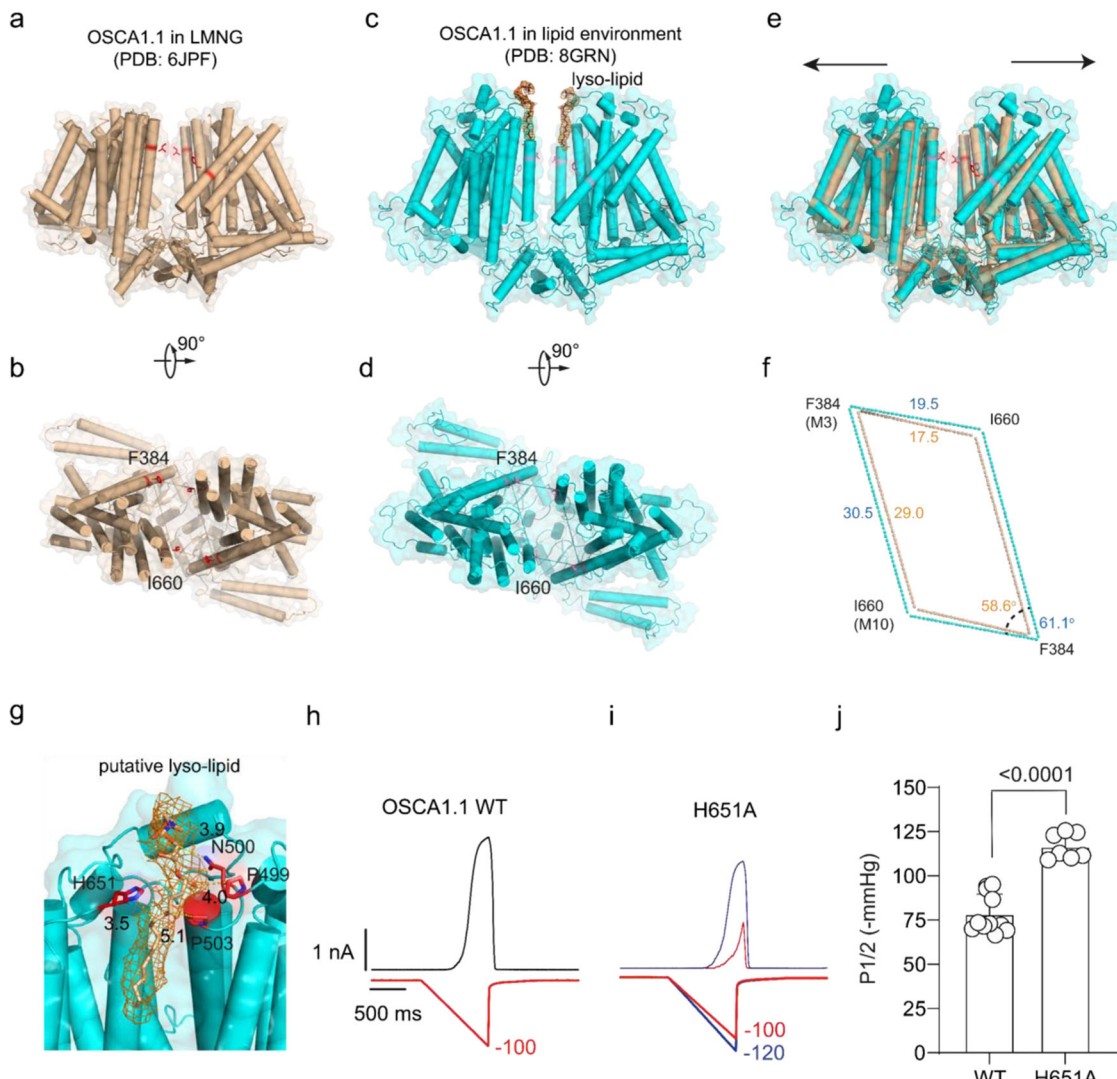

**Fig. 3 | Structure comparison of contracted *AtOSCA*1.1 in a nanodisc environment and extended *AtOSCA*1.1 in a detergent environment. a, b** Cartoon and surface representation of *AtOSCA*1.1 in detergent environment shown in brown, viewed from side (**a**) and top (**b**). The residues F384 on M3 and I660 on M10 are colored in red. **c, d** Cartoon and surface representation of extended state *AtOSCA*1.1 shown in cyan in nanodisc environment, viewed from side (**c**) and top (**d**). **e** Superimposition of the contracted *AtOSCA*1.1 in a detergent environment and extended *AtOSCA*1.1 in a nanodisc environment, viewed from the side. The direction of extension of *AtOSCA*1.1 from the contracted state to the extended state is illustrated with two black arrows. **f** Superimposed residues (F384 and I660) of the

contracted and extended *AtOSCA*1.1 structures. The residues are linked by brown lines in contracted state and cyan lines in extended state. The distance between the specified residues and the angle of the specified lines are shown in Å. **g** Cartoon shows the putative residues including H651 that coordinate the lyso-lipid (the map contour σ = 4.0). Distances are shown in Å. **h, i** Representative traces of pressure activated wild-type (WT) *AtOSCA*1.1 (**h**) and H651A mutation (**i**) currents. **j** Quantification of the half activation pressure (P1/2) of WT *AtOSCA*1.1 and H651A mutation. Data is shown as mean +/− SEM and tested by Student's t-test ($n > 8$ and $p < 0.01$).

elements in *AtOSCA*1.1 channel[13]. In TMEM16A channels, M6 is the most important channel gating element regulated by calcium[23,24]. In addition, the calcium binding of some TMEM16 proteins generates conformational changes at the dimer interface[19]. Thus, the conformational changes at the interface and M6 motion are coupled in some of the calcium activated TMEM16 proteins. Furthermore, the M0 and M6 directly interact with the putative lipid plug we have identified, linking it directly to membrane forces (i.e., curvature or tension). Therefore, we suggest that force applied to the membrane pulls the lipid plug out of the cytosolic pore region which is ultimately coupled to conformational changes in M0 and M6, enabling channel activation.

## Mechanical coupling in OSCA channel subunits

The dimeric architecture of OSCA channels is widely conserved in many protein families (e.g., CLC, TMEM16), which contain both channels and

transporters[23,38]. The monomeric CLC channel, despite not being physiologically relevant, is still functional as a channel and strongly supports the idea that monomers form functional units from these unrelated protein families[23,39]. The mammalian TMEM63 homologs of OSCA channels from our structural insights are also monomers suggestive of a convergent rule within these families where each monomer may act as a separable functional unit. This is supported by the fact that TMEM63A/B still function as MS channels[12]. We have provided strong evidence that the dimer interface formed by the special electrostatic and hydrogen bonding networks in OSCA channels is not conserved in mammalian TMEM63 homologs. Specifically, we show that two TMEM63 homologs from different species purified with different detergents seem to exist as monomers. Despite this strong evidence and the clear fact that the M10 helix in our TMEM63 structures rotates into the space usually occupied by the OSCA channel central cavity, a

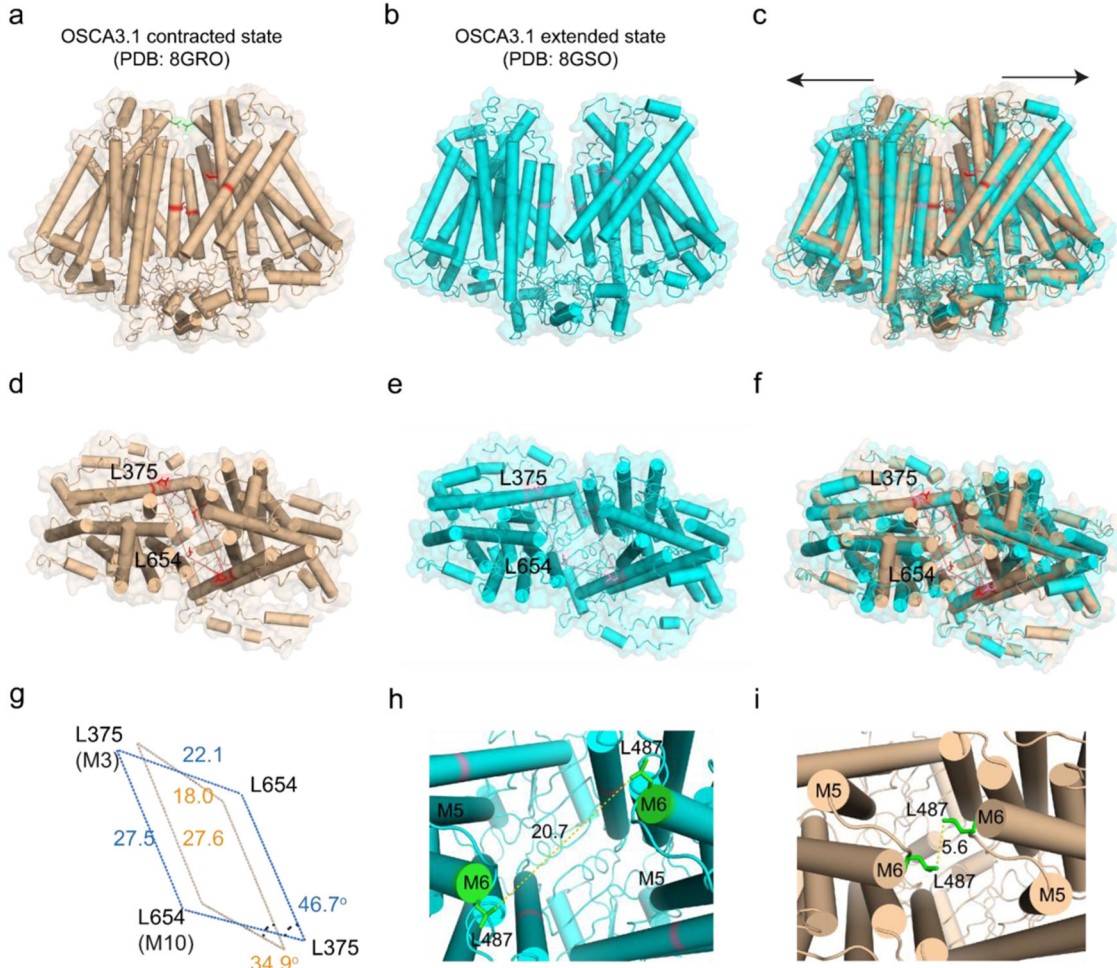

**Fig. 4 | Structure comparison of the contracted and extended *AtOSCA*3.1 states.**
**a–f** Cartoon and surface representation of the contracted *AtOSCA*3.1 (colored in wheat) (**a**) and the extended *AtOSCA*3.1 (colored in cyan) (**b**), viewed from the side, and the contracted *AtOSCA*3.1 (**d**) and the extended *AtOSCA*3.1 (**e**), viewed from the top (in a detergent environment). Superimposition of the contracted and extended *AtOSCA*3.1 structures, viewed from side (**c**) and top (**f**). The extended direction of *AtOSCA*3.1 from contracted state to extended state is shown as two black arrows. **g** Superimposed residues (L375 on M3 and L654 on M10) of the contracted and extended *AtOSCA*1.1 structures. The residues are linked by brown lines in the contracted state and cyan lines in the extended state. The distance between the specified residues and the angle of the specified lines are shown. **h, i** The distance of residues L487 on M5 and M6 linker in the extended state is around 20.7 Å (**h**), while the distance of residues L487 on M5 and M6 linker in the contracted state is around 5.6 Å (**i**). Thus, the residues L487 form an additional dimer interface in the contracted state.

fact that would preclude a similar dimeric organization of TMEM63, we of course cannot rule out that in a native lipid or cellular environment that TMEM63 channels organize as dimers with an alternate dimeric interfacial organization. Indeed, on this point it will be interesting to see whether alternate structural approaches including nanodisc reconstitution of TMEM63 homologs reveal similar monomeric organization.

Furthermore, the monomeric organization of TMEM63A/B also suggests that the central cavity formed in dimeric OSCA channels may not be the primary force sensing element. We identified that the cytosolic pore region is blocked by a putative lipid that could act like a plug (in the same way proposed for protein components in PIEZO1 channel gating[40]), which may be the major energy barrier for channel activation. However, the dimer formation seems to sensitize OSCA channels to mechanical force, therefore force sensing and transduction by dimeric OSCA channels seem to be coupled between the subunits. Our results suggested that the central cavity likely facilitates force transduction through the lipids stored within it as well as the interactions of each subunit. Moreover, when force is applied to each subunit, the central cavity may be a cache for a force feedback response. We refer to this hypothesis as the mechanical-coupling mechanism (Fig. 5a–c). The exact reasons for the oligomeric

differences between plant and mammalian OSCA/TMEM63 homologs are of significant interest and our structures provide a framework to understand how the force-from-lipids paradigm applies at the molecular level to OSCA/TMEM63 channels.

## Methods

### Cell lines
Sf9 cells were from Thermo Fisher Scientific and cultured in Sf-900 II SFM medium (Gibco) at 27 °C. Expi293F suspension cells were from Thermo Fisher Scientific and cultured in HEK293 medium (yocon) supplemented with 1% FBS at 37 °C with 6% $CO_2$ and 70% humidity. Cells were routinely tested for mycoplasma contamination and were negative.

### Fluorescence-detection size-exclusion chromatography
The cDNA for *AtOSCA*1.1 (UniprotKB Q9XEA1), *AtOSCA*3.1 (UniprotKB Q9C8G5), *h*TMEM63A (UniprotKB O94886), and *m*TMEM63B (UniprotKB Q5T3F8) was cloned into a modified C-terminally GFP-tagged BacMam expression vector, which also contains a His$_8$ tag and a FLAG tag after GFP. Expi293F cells transfected with *AtOSCA*1.1-CGFP, *AtOSCA*3.1-CGFP, *h*TMEM63A-CGFP and *m*TMEM63B-CGFP plasmids

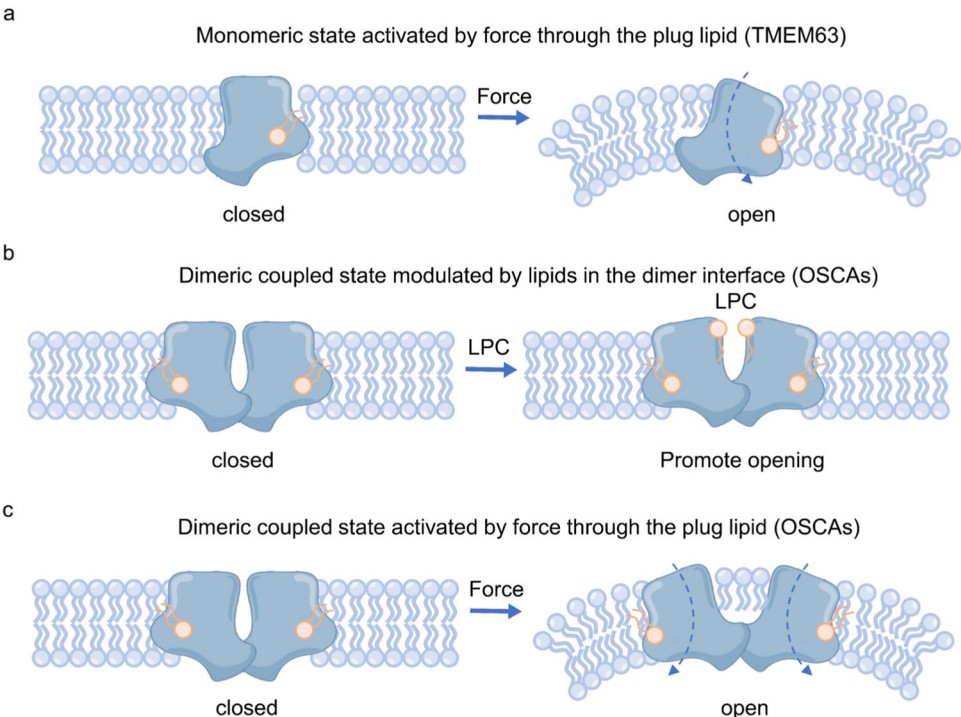

**Fig. 5 | The putative mechanical-coupling mechanism of OSCA/TMEM63 channels. a–c** The putative force-activation model of monomeric TMEM63s or dimeric coupled OSCAs (**a**), the putative lyso-PC-modulation model of dimeric coupled OSCA channels (**b**) and the putative force-activation model of dimeric coupled OSCA channels (**c**) are shown as cartoons. The lipid plug is shown as a wheat cartoon. Force or high-osmolality shock causes deformation of the local membrane, which may induce a pulling force on the lipid plug to drive channel opening (**a**, **c**). The insertion of two putative lyso-PC molecules causes the extension of the central cavity, providing a pathway to channel activation (**b**).

were harvested and solubilized in Tris-buffered saline (TBS; 20 mM Tris pH 8.0 and 150 mM NaCl), 1% (w/v) Lauryl maltose neopentyl glycol (LMNG), and 0.1% (w/v) cholesteryl hemisuccinate (CHS) for 0.5 h at 4 °C, then centrifuged at $15,000 \times g$ to remove insoluble debris. Supernatants were centrifuged at 40,000 r.p.m./$44,800 \times g$ in a TLA55 rotor for 30 min to remove insoluble membrane remnants. Supernatants were injected onto a Superose 6 increase 5/150 column (GE Healthcare), pre-equilibrated with TBS and 10 µM LMNG, and detected by a fluorescence detector (excitation 488 nm and emission 520 nm for GFP signal, excitation 280 nm and emission 335 nm for tryptophan signal). To test how different detergents affected the oligomeric state of $h$TMEM63A proteins, we only replaced the LMNG-CHS with the indicated detergent while all other conditions remained the same.

**Protein expression and purification**
Baculovirus was generated from the DH10Bac bacterial strain and sf9 cells according to a standard Bac-to-Bac protocol. For large-scale expression, Expi293F cells (grown in Yocon HEK293 medium with 6% $CO_2$ at 37 °C and 130 r.p.m.) in suspension were grown to a density of $3.0 \times 10^6$ cells/ml and then infected by baculovirus. 10 mM sodium butyrate was added 12 h post-infection, and the temperature was lowered to 30 °C for protein expression. Cells were harvested 72 h after infection by centrifugation at 4000 r.p.m., 4 °C for 10 min and broken by sonication in cold lysis buffer (20 mM Tris-HCl pH 7.4, 150 mM NaCl). Unbroken cells and cell debris were removed by centrifugation at 8000 r.p.m. for 10 min. The supernatant was centrifuged at 36,000 r.p.m/$49,300 \times g$. for 30 min in a Ti45 rotor (Beckman). Membrane pellets were harvested and frozen at −80 °C until use.

For purification of $h$TMEM63A and $m$TMEM63B in detergent environment, membrane pellets were homogenized in TBS and then solubilized in TBS, 1% (w/v) LMNG, and 0.1% (w/v) CHS for 1 h at 4 °C.

Insoluble material was removed by centrifugation at 40,000 r.p.m./$64,000 \times g$ for 30 min in a Ti45 rotor. The supernatant was loaded onto anti-FLAG G1 affinity resin (GenScript) by gravity flow. Resin was further washed with 10 column volumes of wash buffer (TBS and 0.02% (w/v) LMNG), and protein was eluted with an elution buffer (TBS, 0.02% (w/v) LMNG and 230 µg/ml FLAG peptide). The C-terminal GFP tag of eluted protein was removed by HRV3C protease cleavage for 3 h at 4 °C. The protein was further concentrated by a 50-kDa cutoff concentrator (Millipore) and loaded onto a Superose 6 increase 10/300 column (GE Healthcare) running in TBS and 0.03% (w/v) digitonin. Peak fractions were combined and concentrated to around 9 mg/ml of $h$TMEM63A and 5 mg/ml of $m$TMEM63B for cryo-EM sample preparation.

The membrane scaffold protein MSPE3D1 with N-terminal 6×His tag was expressed in *E. coli* BL21(DE3) cells. Collected cells were lysed by sonication in TBS supplemented with 2 mM phenylmethylsulfonyl fluoride (PMSF). After centrifugation at $20,000 \times g$ for 1 h at 4 °C, the supernatant was loaded onto the nickel-affinity resin. The resin was washed with TBS plus 20 mM imidazole. Protein was eluted with TBS plus 300 mM imidazole. The eluate was concentrated and loaded on a Superdex 200 Increase column equilibrated with TBS. The MSPE3D1-containing fractions were pooled for nanodisc reconstitution.

For purification of *AtOSCA*1.1 and *AtOSCA*3.1, membrane pellets were homogenized in TBS and then solubilized in TBS supplemented with 1% (w/v) n-dodecyl β-D-maltoside (DDM) for 1 h at 4 °C. Insoluble materials were removed by centrifugation at 40,000 r.p.m./$64,000 \times g$ for 30 min in a Ti45 rotor. Supernatant was loaded onto TALON resin (Clontech) by gravity flow. Resin was further washed with 10 column volumes of wash buffer (TBS, 0.02% (w/v) LMNG and 10 mM imidazole), and protein was eluted with an elution buffer (TBS, 0.02% (w/v) LMNG and 250 mM imidazole). The eluate was collected for nanodisc reconstitution.

## Reconstitution of *At*OSCA1.1 into lipid nanodiscs

Lecithin solubilized in chloroform was dried under nitrogen gas and resuspended with 0.7% (w/v) DDM. *At*OSCA1.1, MSPE3D1, and lipid mixture were mixed at a molar ratio of 1:10:500 and incubated at 4 °C for 3 h. Detergents were removed by incubation with Bio-beads SM2 (Bio-Rad) overnight at 4 °C. The protein-lipid mixture was loaded onto anti-FLAG G1 affinity resin equilibrated with TBS. The resin was further washed with 10 column volumes of TBS and protein was eluted with an elution buffer (TBS plus 230 μg/ml FLAG peptide). The C-terminal GFP tag of eluted protein was removed by HRV3C protease cleavage for 3 h at 4 °C. The protein was incubated with TBS further concentrated by a 100-kDa cutoff concentrator (Millipore) and loaded onto a Superose 6 increase 10/300 column (GE Healthcare) running in TBS. Peak fractions were combined and concentrated to around 3 mg/ml of *At*OSCA1.1 for cryo-EM sample preparation.

## EM sample preparation

For cryo-EM sample preparation, aliquots (3 μl) of the protein sample were loaded onto glow-discharged (30 s, 15 mA; Pelco easiGlow, Ted Pella) Au grids (Quantifoil, R1.2/1.3, 300 mesh). The grids were blotted for 4 s with 2 forces after waiting for 5 s and immersed in liquid ethane using Vitrobot (Mark IV, Thermo Fisher Scientific/FEI) in the condition of 100% humidity and 8 °C.

## Data collection

Cryo-EM data were collected on a Titan Krios microscope (FEI) equipped with a cesium corrector operated at 300 kV. For *h*TMEM63A and *m*TMEM63B in detergent micelles, movie stacks were automatically acquired with EPU software on a Gatan K3 Summit detector in super-resolution mode (×105,000 magnification) with pixel size 0.41 Å at the object plane and with defocus ranging from −1.5 μm to −2.0 μm and GIF Quantum energy filter with a 20-eV slit width. The dose rate on the sample was 22.73 $e^-$ $s^{-1}$ $Å^{-2}$, and each stack was 2.20 s long and dose-fractioned into 32 frames with 68.75 ms for each frame. Total exposure was 50 $e^-$ $Å^{-2}$. For *At*OSCA1.1 in lipid nanodiscs and *At*OSCA3.1, movie stacks were automatically acquired with EPU software on a Gatan K3 Summit detector in super-resolution mode (×105,000 magnification) with pixel size 0.42 Å at the object plane and with defocus ranging from −1.5 μm to −2.0 μm and GIF Quantum energy filter with a 20-eV slit width. The dose rate on the sample was 23.92 $e^-$ $s^{-1}$ $Å^{-2}$, and each stack was 2.09 s long and dose-fractioned into 32 frames with 65.31 ms for each frame. Total exposure was 50 $e^-$ $Å^{-2}$.

## Image processing and model building

Data processing was carried out with cryoSPARC v3 suite. Patch CTF estimation was carried out after alignment and summary of all 32 frames in each stack using the patch motion correction. Initial particles were picked from few micrographs using blob picker in cryoSPARC and 2D averages were generated. Final particle picking was done by template picker using templates from those 2D results. Particles were extracted using a box size of 320 pixels. After three rounds of 2D classification, ab initio reconstruction, and non-uniform refinement and local refinement for reconstructing the density map. All maps were low-pass filtered to the map-model FSC value. The reported resolutions were based on the FSC = 0.143 criterion. An initial model was generated by the monomer structure of *At*OSCA1.1. Then, we manually completed and refined the model using Coot. Subsequently, the models were refined against the corresponding maps by PHENIX. We used PyMol and Chimera for structural analysis and generation of graphics. For *h*TMEM63A, we modeled the putative plug lipid using POPC while for AtOSCA1.1 we modeled the central cavity lipid as a lyso-phosphatidylcholine.

## Electrophysiology

HEK293T-Piezo1KO cells were transfected with *At*OSCA1.1-CGFP or mutations and incubated for 24–36 h before recording. Inside-out patches were performed as previously described[13]. Currents were recorded at −20 mV using an Axopatch 200B amplifier at a sampling rate of 10 kHz and filtered at 2 kHz (Digidata 1440A, Molecular Devices). Negative pressure was applied to patch pipettes using a High-Speed Pressure Clamp-1 (ALA Scientific Instruments, Farmingdale, NY, USA) and recorded in millimeters of mercury (mmHg) using a piezo-electric pressure transducer (WPI, Sarasota, FL, USA). We used an I/Imax versus negative pressure curve approach for evaluating mechanosensitivity, where Imax is the maximum current measured from the patch excised in the inside-out patch clamp setting. All recordings were performed at room temperature (22 °C). Data were then analyzed by pClamp10.7 software. All data were acquired from at least three independent cells.

## Reporting summary

Further information on research design is available in the Nature Portfolio Reporting Summary linked to this article.

## Data availability

The data that support this study are available from the corresponding authors upon request. The cryo-EM maps have been deposited in the Electron Microscopy Data Bank (EMDB) under accession codes EMD-34209 (extended state of *At*OSCA1.1), EMD-34210 (contracted state of *At*OSCA3.1), EMD-34237 (extended state of *At*OSCA3.1), and EMD-34214 (*h*TMEM63A in detergent). The atomic coordinates have been deposited in the Protein Data Bank (PDB) under accession codes 8GRN extended state of *At*OSCA1.1, 8GRO (contracted state of *At*OSCA3.1), 8GSO (extended state of *At*OSCA3.1), and 8GRS (*h*TMEM63A in detergent). Structural models used to initiate model building were accessed from the Protein Data Bank under the accession code 6JPF. Source data are provided with this paper.

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

## Acknowledgements

We would like to thank the Cryo-EM Facility and High Performance Computing (HPC) Center of Westlake University for providing cryo-EM and computation support. This work was supported by Westlake Laboratory (Westlake Laboratory of Life Sciences and Biomedicine), an Institutional Startup Grant from the Westlake Education Foundation and National Natural Science Foundation of China (31830060, 92068201) to D.P. We also would like to thank all the Cell fate control lab members for their support.

## Author contributions

M.Z. and D.P. conceived the project. Y.S., C.C., and M.Z. designed the experiments. Y.S. prepared the cryo-EM sample. Y.S. and M.Z. collected cryo-EM data. M.Z. performed image processing and analyzed EM data. Y.S. and M.Z. built the model and wrote the manuscript draft. C.C. performed the electrophysiological studies. All authors contributed to the manuscript preparation.

## Competing interests

The authors declare no competing interests.
