## [Peer Review File · Nature Communications]

A mechanical-coupling mechanism in OSCA channel mechanosensitivityReviewers' Comments:

Reviewer #1:

Remarks to the Author:

Reviewer Assessment

In manuscript#: NCOMMS-22-42667-T titled 'A mechanical-coupling mechanism in OSCA channel mechanosensitivity,' Duanqing Pei and colleagues report new Cryo-EM structures of mammalian homologs of OSCA channels hTMEM63A and AtOSCA channels and propose a lipid-mediated mechanical-coupling mechanism, distinct from plant OSCA mechanosensitive channels, the mammalian homolog hTMEM63A and mTMEM63B exist as monomer instead of the dimer. This is a very novel and exciting result.

The experimental structure determination is excellent. Experimental procedures were given in detail. Experimental results were rigorously analyzed and well presented. However, the mechanosensitive-coupling mechanism proposed is less convincing. The authors compared hTMEM63A and AtOSCA channels with MscS-like mechanosensitive channels. However, the pore lipids in the MscS channel do not directly interact with the cell membrane lipids; the hook lipids directly interact with the cell membrane lipids and mediate the opening and gating of the MscS channel. Although in Fig. 5, the authors made cartoons to show plug lipids potentially interact with cell membranes, there is no convincing solid structural evidence to support this.

Furthermore, the lipids observed in the nanodisc-prepared samples are artificial lyso-lipids. This protein-lipid interaction might be different from the native lipid-protein interaction. It is unclear how the pore lipid in the OSCA channel and how the lipids between the OSCA dimer mediate the gating of these channels.

The discovery of the monomeric mammalian OSCA homolog channel prepared with detergents is interesting; however, it should be cautious when interpreting the structural results obtained from detergents because sometimes detergents could dissociate oligomers. Some detergent-free methods might better preserve the oligomeric state of membrane proteins. If the authors have access to the approach, it is highly recommended to test it. Otherwise, it needs to be mentioned that the monomeric channel might be biased by the detergent-based approach in this study, considering the structural similarity between the plant and mammalian OSCS-like channels.

Overall the manuscript was well written. There are some minor issues: for Fig. 2, in the context, the conserved residues F555, F556 and Y559 on TM6 might be mislabeled. In the figure legend, '...F516, F517 and Y520 on M7 that are blocking...' looks like should be "...on TM6..." In the figure, the transmembrane helices are labeled as M0, M3, M4, M6, and M7; however, in the legend, it is sometimes described as TMx, and sometimes as Mx. The label should be consistent.

There are no particular ethical concerns for this work. The authors have made a significant and novel contribution to the field of mechanosensitive channels. This manuscript merits consideration for publication in Nature Communications as an original research Article.

The reviewer recommends that this manuscript could be published after addressing the major and minor concerns.

Reviewer #2:

Remarks to the Author:

The manuscript by M. Zhang et al examines a hypothetical mechanism of OSCA channels mechanosensitivity based on cryo-EM structures obtained for AtOSCA1.1, AtOSCA3.1 and closely related hTMEM63a and mTMEM63b proteins. First, the authors present hTMEM63a and mTMEM63b structures in detergent. The both proteins purify as monomers, which suggests functional significance of their monomeric forms. For hTMEM63a the authors identify lipid-like density, which, as they hypothesize, may represent a lipid molecule "plugging" the conductive pore of the channel. In AtOSCA1.1 (extended, but presumably closed) structure in nanodiscs the authors identify electron densities that they attribute to lyso-lipids bound inside the central cavity at the dimer interface. The authors further hypothesize that this interaction provides the pathway to a further channel activation.

For less tension-sensitive AtOSCA3.1 structure, obtained in nanodiscs, the authors identified 2 basic (but still closed) structures: compact and extended, the latter being similar to that obtained for AtOSCA1.1.

The data seem to be of sufficient quality, but its presentation is often unclear, not logical and suffers from multiple mistakes. In particular, some parts of the manuscript seem to be poorly related to each other, both scientifically and structurally. Because of that the conclusions of the manuscript are not justified enough. There are quite a few points that require further clarification and rationale as noted below.

1. A bit surprising is that in Summary the authors mention TMEM63 only once in the last sentence, while approximately 1/3 of the manuscript is dedicated to the newly obtained structure of hTMEM6a and its specific details, which supposedly is one of the main points of the paper.

2. In Summary, the authors state: "...we show that the mechanosensitive mammalian homologues of OSCA channels are monomeric channels gated by lipids...", which is not quite correct for the following reasons:

a) the authors show that mammalian TMEM63 purifies as a monomer and that TMEM63s are monomers in detergent micelles. No functional data presented showing the actual oligomeric structure of TMEM63 in cellular membranes or reconstitution systems
b) the authors do not show that (lyso-)lipids actually "gate" TMEM63. Membrane tension does and, as the authors mention later in the text, lipids are likely to modulate (or potentiate) TMEM63. In case of the suggested mechanism, when lipid molecules "plug" the pore, the stimulus is still membrane tension, which pulls the lipid molecule out of the pore.

3. The authors compare AtOSCA and hTMEM63 structures obtained in different systems (lipid nanodiscs and detergent micelles respectively). Any assumptions about channel function, based on such comparison, should be given very cautiously, because:

a) the both systems (esp. micelles) do not recapitulate natural lipid environment of cellular membranes
b) even the same protein in different systems (micelles vs lipid nanodiscs) may adopt very different conformations (a good example is human/mouse TRPV3 channel). Moreover, the authors further report "An extended state of AtOSCA1.1 in lipid nanodiscs..." opposed to a "contracted" state in micelles.

4. In Results, the authors state: "...we found, regardless of the detergent used for extraction, that hTMEM63A and mTMEM63B are monomeric mechanosensitive channels..." Again, this does not appear to be correct, because:

a) only LMNG detergent-based protein purification protocol is mentioned in the Methods section. Figure S1b however illustrates FSEC results for TMEM63a with 3 different detergents, but there are no references to it in the Methods. Moreover, Figure S1 is entitled "Purification and cryo-EM data procedure of AtOSCA1.1 in nanodisc", which does not make sense.
b) the authors show that mammalian TMEM63 purifies as a monomer and that TMEM63s are monomers in detergent micelles only, not in nanodiscs
c) the manuscript does not contain any functional data on mammalian TMEM63s, e.g. regarding their mechanosensation or oligomeric structure in any environment except detergent micelles. Generally, the authors often use the term "monomeric hTMEM63A channel" throughout the text, which has not been functionally proven.

5. In Results, the authors state: "...we restrict our below discussion to hTMEM63A..." without specifying the reason. Although the density map of hTMEM63A seems to be better than that of mTMEM63B (Figure S3), the text specifies the opposite and refers to Figure S1, which however contains AtOSCA1.1 data. The authors should fix that, as well as a few similar typos and broken references throughout the text.

6. Related to the previous point - there are two figures entitled almost identically: "Extended Data Fig. 1 | Purification and cryo-EM data processing procedure of AtOSCA1.1 in nanodisc" and "Extended Data Fig. 4 | Cryo-EM data processing procedure of AtOSCA1.1 in nanodisc". This is very confusing.
7. Additionally, the authors do not present any heat maps for local resolution of any proteins, which raises a question about the exact orientation of side chains, for example, of conductive pore-lining helices.
8. The authors further state: "...Structural comparison of the monomeric hTMEM63A and AtOSCA1.1, demonstrates that the M10 of hTMEM63A is rotated into the 'central cavity' and would sterically hinder the dimeric interface (Extended Data Fig. 3d-e), providing a likely reason for the monomeric form." It is very difficult to understand from the figure why it is the case and which non-conserved residues from the dimer interface could be responsible for that. Again, could not that be an artifact of detergent micelle vs lipid nanodisc systems?
9. Figure 1 lacks "thin black lines" of the "approximate extent of the phospholipid bilayer" or at least I can not find it.
10. Figure 2. The lipid density presented on 2a is very difficult to see (also no σ is reported). On 2f, which is supposed to show the lipid-protein interactions, it is difficult to identify (electrostatically) interacting residues, responsible to lipid molecule docking to this particular location.
11. An extended state of AtOSCA1.1 in lipid nanodiscs is driven by lyso-lipids. The mechanism of "promotion" of OSCA channel activation, suggested in this paragraph, is not quite clear. The authors suggest that lyso-lipids, bound within the central cavity, facilitate gating. Following this logic, hydrophobic mutation H651A (as shown on Figure 3g) should theoretically increase interaction with a lyso-lipid hydrophobic tail to stabilize the extended conformation and, therefore, to facilitate gating even more. Patch-clamp data (Figure 3h-j), however, clearly indicates the opposite - namely, lower tension sensitivity. Could the authors elaborate more on that?
12. Figure 4 does not seem to illustrate the results and conclusions presented in the text. Figures 4a and 4b represent contracted and extended states of OSCA3.1, while Figures 4c and 4d are superposition of 4a and 4b viewed from different angles. Residues labeled in red probably refer to the central cavity cross-section illustrated on Figure 4F, but there are no references to them in the legend, as there is no legend for Figure 4e. The authors state, that single monomers from the both (contracted and expanded) states superimpose and are basically identical, but there is no figure showing that. The interaction between L487 (or L478 as mentioned in the legend?) residues from TM5-TM6 linkers (shown on Figure 4e) is not elaborated in the legend. In particular, it is not clear which state is illustrated, as no distance between these residues is labeled. No differences between leucines interaction in contracted and extended states are illustrated either.
13. Maybe I am missing something, but the presentation of data in the manuscript seems to me quite inconsistent. For example, though the authors discuss hypothetical regulation of OSCA1.1 by lyso-lipids, they do not consider OSCA3.1 or TMEM63. In fact, it is unclear from the text even if any lipid-like densities were detected in OSCA3.1/TMEM63 in the locations, where those were found for OSCA1.1. Vice versa, speaking about hypothetical "gating" of TMEM63 by lipids, the authors completely omit any discussion concerning structurally related OSCA1.1 and OSCA3.1. Unfortunately, such an approach turns the manuscript into a set of poorly related parts.

Point-by-Point Response to Reviewer's Comments

Dear Reviewers:

We would like to express our great gratitude to the editor and reviewers for the valuable comments and suggestions! Our response to review comments was given in this letter and the revised manuscript. In the following, we present our response (marked in blue) to each review comment in detail. We hope the revised manuscript and figures can satisfy your and the reviewers' concerns.

We request your kind consideration of this work for publication in *Nature Communications*.

Reviewer #1's comments:

Assessment: In manuscript#: NCOMMS-22-42667-T titled 'A mechanical-coupling mechanism in OSCA channel mechanosensitivity', Duanqing Pei and colleagues report new Cryo-EM structures of mammalian homologs of OSCA channels hTMEM63A and *At*OSCA channels and propose a lipid-mediated mechanical-coupling mechanism, distinct from plant OSCA mechanosensitive channels, the mammalian homolog hTMEM63A and mTMEM63B exist as monomer instead of the dimer. This is a very novel and exciting result.

Response: We appreciate the reviewer for the encouragement and positive comments.

Comment 1, The experimental structure determination is excellent. Experimental procedures were given in detail. Experimental results were rigorously analyzed and well presented. However, the mechanosensitive-coupling mechanism proposed is less convincing. The authors compared hTMEM63A and *At*OSCA channels with MscS-like mechanosensitive channels. However, the pore lipids in the MscS channel do not directly interact with the cell membrane lipids; the hook lipids directly interact with the cell membrane lipids and mediate the opening and gating of the MscS channel. Although in Fig. 5, the authors made cartoons to show plug lipids potentially interact with cell membranes, there is no convincing solid structural evidence to support this. Furthermore, the lipids observed in the nanodisc-prepared samples are artificial lyso-lipids. This protein-lipid interaction might be different from the native lipid-protein interaction. It is unclear how the pore lipid in the OSCA channel and how the lipids between the OSCA dimer mediate the gating of these channels.

Response: We appreciate the reviewer's constructive suggestion. We revised the model figure (Fig. 5) as the reviewer's suggestion and tune down (using the putative and potential etc.) the statement of the lipid especially the lyso-lipids modulated the channel gating.

Comment 2, The discovery of the monomeric mammalian OSCA homolog channel prepared with detergents is interesting; however, it should be cautious when interpreting

the structural results obtained from detergents because sometimes detergents could dissociate oligomers. Some detergent-free methods might better preserve the oligomeric state of membrane proteins. If the authors have access to the approach, it is highly recommended to test it. Otherwise, it needs to be mentioned that the monomeric channel might be biased by the detergent-based approach in this study, considering the structural similarity between the plant and mammalian OSCS-like channels.

Response: Thanks for this constructive suggestion. We mentioned the monomeric channel might be biased by the detergent-based approach in the discussion.

Minor comments,

1. for Fig. 2, in the context, the conserved residues F555, F556 and Y559 on TM6 might be mislabeled. In the figure legend, ‘...F516, F517 and Y520 on M7 that are blocking...’ looks like should be “...on TM6...”.
2. In the figure, the transmembrane helices are labeled as M0, M3, M4, M6, and M7; however, in the legend, it is sometimes described as TMx, and sometimes as Mx. The label should be consistent.

Response: We are very grateful for such detailed modifications to our manuscript. We have revised these mistakes one by one according to the reviewer’s suggestion.

Reviewer 2’s comments:

Assessment: The manuscript by M. Zhang et al examines a hypothetical mechanism of OSCA channels mechanosensitivity based on cryo-EM structures obtained for AtOSCA1.1, AtOSCA3.1 and closely related hTMEM63a and mTMEM63b proteins. First, the authors present hTMEM63a and mTMEM63b structures in detergent. The both proteins purify as monomers, which suggests functional significance of their monomeric forms. For hTMEM63a the authors identify lipid-like density, which, as they hypothesize, may represent a lipid molecule “plugging” the conductive pore of the channel. In AtOSCA1.1 (extended, but presumably closed) structure in nanodiscs the authors identify electron densities that they attribute to lyso-lipids bound inside the central cavity at the dimer interface. The authors further hypothesize that this

interaction provides the pathway to a further channel activation. For less tension-sensitive AtOSCA3.1 structure, obtained in nanodiscs, the authors identified 2 basic (but still closed) structures: compact and extended, the latter being similar to that obtained for AtOSCA1.1.

The data seem to be of sufficient quality, but its presentation is often unclear, not logical and suffers from multiple mistakes. In particular, some parts of the manuscript seem to be poorly related to each other, both scientifically and structurally. Because of that the conclusions of the manuscript are not justified enough. There are quite a few points that require further clarification and rationale as noted below.

Response: We are thankful to the reviewer for the comments, which greatly helped us to improve the quality of our manuscript. We have carefully considered each criticism and made appropriate modifications.

Comment 1, A bit surprising is that in Summary the authors mention TMEM63 only once in the last sentence, while approximately 1/3 of the manuscript is dedicated to the newly obtained structure of hTMEM63a and its specific details, which supposedly is one of the main points of the paper.

Response: Thanks for the reviewer's suggestion. We used OSCA/TMEM63 throughout the paper.

Comment 2, In Summary, the authors state: "...we show that the mechanosensitive mammalian homologues of OSCA channels are monomeric channels gated by lipids...", which is not quite correct for the following reasons:

a) the authors show that mammalian TMEM63 purifies as a monomer and that TMEM63s are monomers in detergent micelles. No functional data presented showing the actual oligomeric structure of TMEM63 in cellular membranes or reconstitution systems

b) the authors do not show that (lyso-)lipids actually "gate" TMEM63. Membrane tension does and, as the authors mention later in the text, lipids are likely to modulate (or potentiate) TMEM63. In case of the suggested mechanism, when lipid molecules

“plug” the pore, the stimulus is still membrane tension, which pulls the lipid molecule out of the pore.

Response: We thank the reviewer’s comment. We attempted to reconstitute the human TMEM63A in GUV, but as the human TMEM63A is too stiff like some MscL channels in bacteria, we cannot successfully obtain the reconstituted MS current. However, we mentioned the monomeric channel might be biased by the detergent-based approach in the discussion as reviewer #1’s constructive suggestion. Meanwhile, we also toned down (using the putative and potential etc.) the statement of the lipid especially the lyso-lipids modulated the channel gating.

Comment 3, The authors compare AtOSCA and hTMEM63 structures obtained in different systems (lipid nanodiscs and detergent micelles respectively). Any assumptions about channel function, based on such comparison, should be given very cautiously, because:

- a) the both systems (esp. micelles) do not recapitulate natural lipid environment of cellular membranes
- b) even the same protein in different systems (micelles vs lipid nanodiscs) may adopt very different conformations (a good example is human/mouse TRPV3 channel). Moreover, the authors further report “An extended state of AtOSCA1.1 in lipid nanodiscs...” opposed to a “contracted” state in micelles.

Response: Thanks to the review’s comments and we quite agreed with the review’s opinions on different systems that may result in different conformations. However, the contracted and extended states exist in both OSCA1.1 and OSCA3.1 and the contracted and extended states seem largely exist in the mechano-related proteins such as prestin and TMC. Therefore, we pointed out the conserved contracted and extended states in this paper and further sophisticated roles of contracted and extended states in mechano-sensing remain more efforts to uncover.

Comment 4, In Results, the authors state: “...we found, regardless of the detergent used for extraction, that hTMEM63A and mTMEM63B are monomeric mechanosensitive

channels...” Again, this does not appear to be correct, because:

a) only LMNG detergent-based protein purification protocol is mentioned in the Methods section. Figure S1b however illustrates FSEC results for TMEM63a with 3 different detergents, but there are no references to it in the Methods. Moreover, Figure S1 is entitled “Purification and cryo-EM data procedure of AtOSCA1.1 in nanodisc”, which does not make sense

b) the authors show that mammalian TMEM63 purifies as a monomer and that TMEM63s are monomers in detergent micelles only, not in nanodiscs

c) the manuscript does not contain any functional data on mammalian TMEM63s, e.g. regarding their mechanosensation or oligomeric structure in any environment except detergent micelles.

Generally, the authors often use the term “monomeric hTMEM63A channel” throughout the text, which has not been functionally proven.

Response: We appreciated the reviewer’s comment and apology for these mistakes. We added the different detergent extraction protocols in the methods section and revised the Supplementary Fig. 1.

Comment 5, In Results, the authors state: “...we restrict our below discussion to hTMEM63A...” without specifying the reason. Although the density map of hTMEM63A seems to be better than that of mTMEM63B (Figure S3), the text specifies the opposite and refers to Figure S1, which however contains AtOSCA1.1 data. The authors should fix that, as well as a few similar typos and broken references throughout the text. Related to the previous point - there are two figures entitled almost identically: “Extended Data Fig. 1 | Purification and cryo-EM data processing procedure of AtOSCA1.1 in nanodisc” and “Extended Data Fig. 4 | Cryo-EM data processing procedure of AtOSCA1.1 in nanodisc”. This is very confusing.

Response: Thank you for reminding us of the call to Figure and oversighting in our previous manuscript. We have carefully revised these issues in the revised manuscript accordingly, and have re-scrutinized the whole manuscript. We are sorry for the mistake in the figure legend of Supplementary Fig. 1-2. We have fixed it in the revised

manuscript.

Comment 6, Additionally, the authors do not present any heat maps for local resolution of any proteins, which raises a question about the exact orientation of side chains, for example, of conductive pore-lining helices.

Response: We have added the heat maps for local resolution of all proteins in supplementary Figures.

Comment 7, The authors further state: “...Structural comparison of the monomeric hTMEM63A and AtOSCA1.1, demonstrates that the M10 of hTMEM63A is rotated into the ‘central cavity’ and would sterically hinder the dimeric interface (Extended Data Fig. 3d-e), providing a likely reason for the monomeric form.” It is very difficult to understand from the figure why it is the case and which non-conserved residues from the dimer interface could be responsible for that. Again, could not that be an artifact of detergent micelle vs lipid nanodisc systems?

Response: Our previously solved conformation of dimeric AtOSCA1.1 in a detergent environment shows the dimer interface of the AtOSCA1.1 is formed by electrostatic and hydrogen bonding networks (Q and T repeats and R-E pairs) and M10 is isolated by lipids (Zhang etc. NSMB 25, 850–858 (2018)). However, the mammalian OSCA/TMEM63s are not conserved at the electrostatic and hydrogen bonding networks (Q and T repeats and R-E pairs) (Supplementary Fig. 4f). And the structure comparison of dimeric AtOSCA1.1 and monomeric hTMEM63A (Supplementary Fig. 4d-e) shows that there is a rotation of TM10 in hTMEM63A compared to that in AtOSCA1.1, which sterically hinders the formation of the dimer interface. And all structures mentioned above are obtained in a detergent environment. Therefore, it is difficult for mammalian OSCA/TMEM63s to form dimer channels.

Comment 8, Figure 1 lacks “thin black lines” of the “approximate extent of the phospholipid bilayer” or at least I can not find it.

Response: We thank the reviewer’s reminder. The thin blank lines have been added in Figure 1.

Comment 9, Figure 2. The lipid density presented on 2a is very difficult to see (also no σ is reported). On 2f, which is supposed to show the lipid-protein interactions, it is difficult to identify (electrostatically) interacting residues, responsible to lipid molecule docking to this particular location.

Response: We have modified figures 2a and 2f shown below to make the lipid density and lipid binding site clearer.

Comment 10, An extended state of AtOSCA1.1 in lipid nanodiscs is driven by lyso-lipids. The mechanism of “promotion” of OSCA channel activation, suggested in this paragraph, is not quite clear. The authors suggest that lyso-lipids, bound within the central cavity, facilitate gating. Following this logic, hydrophobic mutation H651A (as shown on Figure 3g) should theoretically increase interaction with a lyso-lipid hydrophobic tail to stabilize the extended conformation and, therefore, to facilitate gating even more. Patch-clamp data (Figure 3h-j), however, clearly indicates the opposite – namely, lower tension sensitivity. Could the authors elaborate more on that?

Response: We thank the reviewer’s suggestion. The H651A may impair the putative lyso-lipid and channel interaction thus exhibiting lower tension sensitivity.

Comment 11, Figure 4 does not seem to illustrate the results and conclusions presented in the text. Figures 4a and 4b represent contracted and extended states of OSCA3.1, while Figures 4c and 4d are superposition of 4a and 4b viewed from different angles. Residues labeled in red probably refer to the central cavity cross-section illustrated on Figure 4F, but there are no references to them in the legend, as there is no legend for Figure 4e. The authors state, that single monomers from the both (contracted and expanded) states superimpose and are basically identical, but there is no figure showing that. The interaction between L487 (or L478 as mentioned in the legend?) residues from TM5-TM6 linkers (shown on Figure 4e) is not elaborated in the legend. In particular, it is not clear which state is illustrated, as no distance between these residues is labeled. No differences between leucines interaction in contracted and extended states are illustrated either.

Response: Thanks for this suggestion. We have modified Figure 4 and its legend as the reviewer's suggestion.

Comment 12, Maybe I am missing something, but the presentation of data in the manuscript seems to me quite inconsistent. For example, though the authors discuss hypothetical regulation of OSCA1.1 by lyso-lipids, they do not consider OSCA3.1 or TMEM63. In fact, it is unclear from the text even if any lipid-like densities were detected in OSCA3.1/TMEM63 in the locations, where those were found for OSCA1.1. Vice versa, speaking about hypothetical "gating" of TMEM63 by lipids, the authors completely omit any discussion concerning structurally related OSCA1.1 and OSCA3.1. Unfortunately, such an approach turns the manuscript into a set of poorly related parts.

Response: We appreciated the reviewer's comments. In the corresponding region of hTMEM63a, there is a similar plug-lipid in OSCA1.1 and OSCA3.1 albeit the density is not better than hTMEM63a. We add the potential plug-lipid density in OSCA1.1 and OSCA3.1 in the supplementary figure 8. And we discussed the structurally related hTMEM63a, OSCA1.1 and OSCA3.1 in the first discussion part.

Thanks again to editors and reviewers for putting so much effort into reviewing our paper.

Reviewers' Comments:

Reviewer #2:

Remarks to the Author:

Reviewer 2's comments:

Assessment: The manuscript by M. Zhang et al examines a hypothetical mechanism of OSCA channels mechanosensitivity based on cryo-EM structures obtained for AtOSCA1.1, AtOSCA3.1 and closely related hTMEM63a and mTMEM63b proteins. First, the authors present hTMEM63a and mTMEM63b structures in detergent. The both proteins purify as monomers, which suggests functional significance of their monomeric forms. For hTMEM63a the authors identify lipid-like density, which, as they hypothesize, may represent a lipid molecule "plugging" the conductive pore of the channel. In AtOSCA1.1 (extended, but presumably closed) structure in nanodiscs the authors identify electron densities that they attribute to lyso-lipids bound inside the central cavity at the dimer interface. The authors further hypothesize that this interaction provides the pathway to a further channel activation. For less tension- sensitive AtOSCA3.1 structure, obtained in nanodiscs, the authors identified 2 basic (but still closed) structures: compact and extended, the latter being similar to that obtained for AtOSCA1.1.

The data seem to be of sufficient quality, but its presentation is often unclear, not logical and suffers from multiple mistakes. In particular, some parts of the manuscript seem to be poorly related to each other, both scientifically and structurally. Because of that the conclusions of the manuscript are not justified enough. There are quite a few points that require further clarification and rationale as noted below.

Response: We are thankful to the reviewer for the comments, which greatly helped us to improve the quality of our manuscript. We have carefully considered each criticism and made appropriate modifications.

Reviewer: I appreciate the authors' efforts, however many alterations introduced to the newest version of the manuscript appear to be cosmetic and do not address my concerns. In addition to that, there are still multiple issues with figures and their legends that are considered in detail below.

Comment 1, A bit surprising is that in Summary the authors mention TMEM63 only once in the last sentence, while approximately 1/3 of the manuscript is dedicated to the newly obtained structure of hTMEM63a and its specific details, which supposedly is one of the main points of the paper.

Response: Thanks for the reviewer's suggestion. We used OSCA/TMEM63 throughout the paper.

Reviewer: OK.

Comment 2, In Summary, the authors state: "...we show that the mechanosensitive mammalian homologues of OSCA channels are monomeric channels gated by lipids...", which is not quite correct for the following reasons:

- a) the authors show that mammalian TMEM63 purifies as a monomer and that TMEM63s are monomers in detergent micelles. No functional data presented showing the actual oligomeric structure of TMEM63 in cellular membranes or reconstitution systems
- b) the authors do not show that (lyso-)lipids actually "gate" TMEM63. Membrane tension does and, as the authors mention later in the text, lipids are likely to modulate (or potentiate) TMEM63. In case of the suggested mechanism, when lipid molecules "plug" the pore, the stimulus is still membrane tension, which pulls the lipid molecule out of the pore.

Response: We thank the reviewer's comment. We attempted to reconstitute the human TMEM63A in GUV, but as the human TMEM63A is too stiff like some MscL channels in bacteria, we cannot successfully obtain the reconstituted MS current. However, we mentioned the monomeric channel might be biased by the detergent-based approach in the discussion as reviewer #1's constructive suggestion. Meanwhile, we also toned down (using the putative and potential etc.) the statement of the lipid especially the lyso-lipids modulated the channel gating.

Reviewer: Unfortunately, looking at the updated manuscript, I do not see the alterations described above. For example, right in the Summary, lines 20-21, the authors state: "Here we show that the mechanosensitive mammalian homologues of OSCA/TMEM63 channels are monomeric channels gated by lipids...". In Results, lines 77-78: "...we found, regardless of the detergent used for extraction, that hTMEM63A 78 and mTMEM63B are monomeric mechanosensitive channels". As I mentioned before, no experimental data, demonstrating functional relevance of monomeric TMEM63, are presented. Also, gating of TMEM63 by lipids is still a hypothesis, not an established fact.

Comment 3, The authors compare AtOSCA and hTMEM63 structures obtained in different systems (lipid nanodiscs and detergent micelles respectively). Any assumptions about channel function, based on such comparison, should be given very cautiously, because:

- a) the both systems (esp. micelles) do not recapitulate natural lipid environment of cellular membranes
- b) even the same protein in different systems (micelles vs lipid nanodiscs) may adopt very different conformations (a good example is human/mouse TRPV3 channel). Moreover, the authors further report "An extended state of AtOSCA1.1 in lipid nanodiscs..." opposed to a "contracted" state in micelles.

Response: Thanks to the reviewer's comments and we quite agreed with the reviewer's opinions on different systems that may result in different conformations. However, the contracted and extended states exist in both OSCA1.1 and OSCA3.1 and the contracted and extended states seem largely exist in the mechano-related proteins such as prestin and TMC. Therefore, we pointed out the conserved contracted and extended states in this paper and further sophisticated roles of contracted and extended states in mechano-sensing remain more efforts to uncover.

Reviewer: OK.

Comment 4, In Results, the authors state: "...we found, regardless of the detergent used for extraction, that hTMEM63A and mTMEM63B are monomeric mechanosensitive channels..." Again, this does not appear to be correct, because:

- a) only LMNG detergent-based protein purification protocol is mentioned in the Methods section. Figure S1b however illustrates FSEC results for TMEM63a with 3 different detergents, but there are no references to it in the Methods. Moreover, Figure S1 is entitled "Purification and cryo-EM data procedure of AtOSCA1.1 in nanodisc", which does not make sense
- b) the authors show that mammalian TMEM63 purifies as a monomer and that TMEM63s are monomers in detergent micelles only, not in nanodiscs
- c) the manuscript does not contain any functional data on mammalian TMEM63s, e.g. regarding their mechanosensation or oligomeric structure in any environment except detergent micelles. Generally, the authors often use the term "monomeric hTMEM63A channel" throughout the text, which has not been functionally proven.

Response: We appreciated the reviewer's comment and apology for these mistakes. We added the

different detergent extraction protocols in the methods section and revised the Supplementary Fig. 1.

Reviewer: While Figure S1 title has been corrected and the Methods section now contains the appropriate details, the statement about "TMEM63 monomeric mechanosensitive channels" (lines 77-78) remains experimentally unsupported. Figure S1 also contains a number of inconsistencies and lacks important information (see details below).

Comment 5, In Results, the authors state: "...we restrict our below discussion to hTMEM63A..." without specifying the reason. Although the density map of hTMEM63A seems to be better than that of mTMEM63B (Figure S3), the text specifies the opposite and refers to Figure S1, which however contains AtOSCA1.1 data. The authors should fix that, as well as a few similar typos and broken references throughout the text. Related to the previous point - there are two figures entitled almost identically: "Extended Data Fig. 1 | Purification and cryo-EM data processing procedure of AtOSCA1.1 in nanodisc" and "Extended Data Fig. 4 | Cryo-EM data processing procedure of AtOSCA1.1 in nanodisc". This is very confusing.

Response: Thank you for reminding us of the call to Figure and oversighting in our previous manuscript. We have carefully revised these issues in the revised manuscript accordingly, and have re-scrutinized the whole manuscript. We are sorry for the mistake in the figure legend of Supplementary Fig. 1-2. We have fixed it in the revised manuscript.

Reviewer: While the figures titles and references are now (mostly) fixed, there are still multiple typos and errors in the text. Addressing the first part of my comment - lines 82-83 indicate 4Å resolution for hTMEM63A and 3.3Å resolution for mTMEM63B, which is the opposite to the data presented in figures S2 and S3, that report higher resolution of hTMEM63A structure. This is probably the reason why the authors focused on a human protein (although it is not mentioned in the text).

Comment 6, Additionally, the authors do not present any heat maps for local resolution of any proteins, which raises a question about the exact orientation of side chains, for example, of conductive pore-lining helices.

Response: We have added the heat maps for local resolution of all proteins in supplementary Figures.

Reviewer: OK.

Comment 7, The authors further state: "...Structural comparison of the monomeric hTMEM63A and AtOSCA1.1, demonstrates that the M10 of hTMEM63A is rotated into the 'central cavity' and would sterically hinder the dimeric interface (Extended Data Fig. 3d-e), providing a likely reason for the monomeric form." It is very difficult to understand from the figure why it is the case and which non-conserved residues from the dimer interface could be responsible for that. Again, could not that be an artifact of detergent micelle vs lipid nanodisc systems?

Response: Our previously solved conformation of dimeric AtOSCA1.1 in a detergent environment shows the dimer interface of the AtOSCA1.1 is formed by electrostatic and hydrogen bonding networks (Q and T repeats and R-E pairs) and M10 is isolated by lipids (Zhang etc. NSMB 25, 850-858 (2018)). However, the mammalian OSCA/TMEM63s are not conserved at the electrostatic and hydrogen bonding networks (Q and T repeats and R-E pairs) (Supplementary Fig. 4f). And the structure comparison of dimeric AtOSCA1.1 and monomeric hTMEM63A (Supplementary Fig. 4d-e) shows that

there is a rotation of TM10 in hTMEM63A compared to that in AtOSCA1.1, which sterically hinders the formation of the dimer interface. And all structures mentioned above are obtained in a detergent environment. Therefore, it is difficult for mammalian OSCA/TMEM63s to form dimer channels.

Reviewer: Unfortunately it is difficult to agree with the authors. First, the manuscript lacks any discussion or at least a description (similar to the mentioned above) of the interactions maintaining dimeric structure of AtOSCA1.1. Second, although the authors highlight the differences between a dimer interface of AtOSCA1.1 and a corresponding region in hTMEM63A (illustrated in Figure S4), the possibility of forming a different set of interactions between hTMEM63A monomers is not discussed either. In addition to that, Figure S4 itself suffers from a lot of inconsistencies (more details below), which significantly decreases its illustrative value.

Comment 8, Figure 1 lacks "thin black lines" of the "approximate extent of the phospholipid bilayer" or at least I can not find it.

Response: We thank the reviewer's reminder. The thin blank lines have been added in Figure 1.

Reviewer: OK.

Comment 9, Figure 2. The lipid density presented on 2a is very difficult to see (also no σ is reported). On 2f, which is supposed to show the lipid-protein interactions, it is difficult to identify (electrostatically) interacting residues, responsible to lipid molecule docking to this particular location.

Response: We have modified figures 2a and 2f shown below to make the lipid density and lipid binding site clearer.

Reviewer: OK, however there is no information (either in the figure legend or Methods) on what lipid molecule was used for modeling.

Comment 10, An extended state of AtOSCA1.1 in lipid nanodiscs is driven by lyso- lipids. The mechanism of "promotion" of OSCA channel activation, suggested in this paragraph, is not quite clear. The authors suggest that lyso-lipids, bound within the central cavity, facilitate gating. Following this logic, hydrophobic mutation H651A (as shown on Figure 3g) should theoretically increase interaction with a lyso-lipid hydrophobic tail to stabilize the extended conformation and, therefore, to facilitate gating even more. Patch-clamp data (Figure 3h-j), however, clearly indicates the opposite - namely, lower tension sensitivity. Could the authors elaborate more on that?

Response: We thank the reviewer's suggestion. The H651A may impair the putative lyso-lipid and channel interaction thus exhibiting lower tension sensitivity.

Reviewer: The authors call H651 residue a "putative lyso-lipid coordinating residue" (line 141), however, Figure 3G hardly illustrates that statement. Neither interacting (electrostatically in case of ionized histidine side chain or hydrophobic if it is not) residues are labeled, nor the distance between any atoms are shown.

Comment 11, Figure 4 does not seem to illustrate the results and conclusions presented in the text.

Figures 4a and 4b represent contracted and extended states of OSCA3.1, while Figures 4c and 4d are superposition of 4a and 4b viewed from different angles. Residues labeled in red probably refer to the central cavity cross-section illustrated on Figure 4F, but there are no references to them in the legend, as there is no legend for Figure 4e. The authors state, that single monomers from the both (contracted and expanded) states superimpose and are basically identical, but there is no figure showing that. The interaction between L487 (or L478 as mentioned in the legend?) residues from TM5-TM6 linkers (shown on Figure 4e) is not elaborated in the legend. In particular, it is not clear which state is illustrated, as no distance between these residues is labeled. No differences between leucines interaction in contracted and extended states are illustrated either.

Response: Thanks for this suggestion. We have modified Figure 4 and its legend as the reviewer's suggestion.

Reviewer: Unfortunately, though the modifications of Figure 4 addressed most of concerns, there are still several issues with this figure (more details below).

Comment 12, Maybe I am missing something, but the presentation of data in the manuscript seems to me quite inconsistent. For example, though the authors discuss hypothetical regulation of OSCA1.1 by lyso-lipids, they do not consider OSCA3.1 or TMEM63. In fact, it is unclear from the text even if any lipid-like densities were detected in OSCA3.1/TMEM63 in the locations, where those were found for OSCA1.1. Vice versa, speaking about hypothetical "gating" of TMEM63 by lipids, the authors completely omit any discussion concerning structurally related OSCA1.1 and OSCA3.1. Unfortunately, such an approach turns the manuscript into a set of poorly related parts.

Response: We appreciated the reviewer's comments. In the corresponding region of hTMEM63a, there is a similar plug-lipid in OSCA1.1 and OSCA3.1 albeit the density is not better than hTMEM63a. We add the potential plug-lipid density in OSCA1.1 and OSCA3.1 in the supplementary figure 8. And we discussed the structurally related hTMEM63a, OSCA1.1 and OSCA3.1 in the first discussion part.

Reviewer: It is not quite clear what the authors mean in lines 176-179. Are the lipid-like densities found in AtOSCA structures weaker than that from hTMEM63A? If so, how significant is the difference? Do authors suggest that the stronger densities imply stronger protein-lipid interactions, which result in decreased tension sensitivity? Hypothesized lipid densities presented in Figure 8 are not color-coded and blend into the protein density, so it is very difficult to compare them.

Miscellaneous:

Lines 100-102: F555, F556, and Y559 all belong to M6 helix, but what are the other pore-lining residues (from other helices)?

Line 125: authors mention AtOSCA1.1 structure obtained in detergent, but do not mention any references or PDB.

Lines 145-150: was AtOSCA3.1 structure obtained also in nanodiscs?

Lines 180-181: I do not understand the logic of the statement about a putative (!) modulating lyso-lipid playing however an important role in the activation of dimeric OSCA/TMEM63 channels. If it is not quite clear yet, if there is a modulating lipid, how one can state that it plays an important role?

After the authors fixed the figure legends references and numbers, there are still multiple errors and inconsistencies in figures and figure legends:

Figure 1. It is not clear, which (extended or contracted, nanodiscs or detergent) conformation of AtOSCA1.1 is compared to hTMEM63A. Would be a good idea to include structures' PDB codes too.

Figure 2. Line 371 – "A lipid that plugs..." - how confident the authors are it could not be a detergent molecule? Line 374 – how was the pore profile calculated – HOLE, anything else? Line 377 (e) – it is not quite clear why the authors only presented M6 side chains, what about other pore-lining helices

(M4, M5)? Line 380 – “The lipid is shown...” - modeled with what kind of lipid (DOPC, etc)? The distances in (f) are Å I suppose.

Figure 3. Line 387 – “a-b” should be “a, d” based on what is presented in the figure. Line 394 – “black arrows”. Line 396 – the distances are in Å? In (g) – could the authors also provide electron densities for the interacting residues/atoms?

Figure 4. The last sentence (color coding description) should go to the top. Are the structures obtained in nanodiscs (as AtOSCA1.1)? Line 407 – “black arrows”. Lines 411-412 – “L478” is mentioned in the legend, but “L487” is labeled in the figure.

Figure 5. Lines 418-419 - “...or dimeric-coupled OSCA/TMEM63 (a)...”, while (a) represents only one monomer.

Figure S1. The graphs lack axis scales (elution volume and fluorescence intensity units). (a) – what detergent was used? Based on the Results text it is probably LMNG? (b) is very confusing: first, black color-coded profile is not annotated (is it DDM, which is supposed to be brown?); second, LMNG profiles for hTMEM63A in (a, brown) and (b, cyan) look different. Would be a good idea to use the same elution volume scale in both (a) and (b) and to mention the column type used. Line 429 – “traces”. Line 431 – “...peak are denoted...”.

Figure S2. Line 442 – “is based”.

Figure S4. Line 460 – “...superposition of them.” (e) – no description of the black dashed arrow, “TM10” in the figure, but “M10” in the legend. (e, f) – do the red boxes represent the same region of M10? If this is a full transmembrane helix, its length in the alignment seems to be quite short – only 10 amino acids. Alignment itself also looks very strange, for example, the conserved cysteine (lower panel, second aa from the left, AtOSCA1.1 residue 342) is not labeled as conserved; several letters (M, D, and W) are clipped (also applies to panel (i)), etc. (i) – “N-terminus”. (g, h) – what are the surfaces presented? What do their colors (blue and brown) mean? (j, k) – is (j) supposed to be AtOSCA1.1 structure (as AtOSCA1.1 residues are mapped onto it)? Also, color coding is inconsistent with (d, e). If it is indeed mTMEM63B (as the legend states), what is the reason for placing it there, as it is discussed neither in the text nor in the figure legend? (l) – erroneously duplicates (i); no description of color coding of the structures. (n) is described as “mTMEM63B” in the legend; there are no comments on the red balls, probably representing bound Ca²⁺ ions; color coding is inconsistent between (m) and (n); the structures are shown from different angles and are also differently scaled, which makes them nearly impossible to compare.

Figure S5. (a) - protein gel ladder is not size-labeled. Line 477 – “black arrows” as far as I can see.

Figure S6. Line 496 – “...is based on...”. A legend for panel (g) is missing.

Figure S7. Line 509 – RMSD (not RSMD) in Å I guess is meant here. What is the color coding in (c)?

Figure S8. I think it would be a very good idea to use a different color for the hypothesized lipid densities in all panels, otherwise they blend with a protein density and are almost impossible to see.

REVIEWER COMMENTS

Reviewer 2's comments:

Reviewer: I appreciate the authors' efforts, however many alterations introduced to the newest version of the manuscript appear to be cosmetic and do not address my concerns. In addition to that, there are still multiple issues with figures and their legends that are considered in detail below.

Response II: We thank the reviewer for the detailed comments and apologize for not fully and completely addressing these in the first round of revisions. We have now extensively edited the figure legends modified the presentation of numerous figures and toned down the claims and extended the discussion.

Comment 2, In Summary, the authors state: "...we show that the mechanosensitive mammalian homologues of OSCA channels are monomeric channels gated by lipids...", which is not quite correct for the following reasons:

- a) the authors show that mammalian TMEM63 purifies as a monomer and that TMEM63s are monomers in detergent micelles. No functional data presented showing the actual oligomeric structure of TMEM63 in cellular membranes or reconstitution systems**
- b) the authors do not show that (lyso-)lipids actually "gate" TMEM63. Membrane tension does and, as the authors mention later in the text, lipids are likely to modulate (or potentiate) TMEM63. In case of the suggested mechanism, when lipid molecules "plug" the pore, the stimulus is still membrane tension, which pulls the lipid molecule out of the pore.**

Response: We thank the reviewer's comment. We attempted to reconstitute the human TMEM63A in GUV, but as the human TMEM63A is too stiff like some MscL channels in bacteria, we cannot successfully obtain the reconstituted MS current. However, we mentioned the monomeric channel might be biased by the detergent-based approach in the discussion as reviewer #1' constructive suggestion. Meanwhile, we also toned down (using the putative and potential etc.) the statement of the lipid especially the lyso-lipids modulated the channel gating.

Reviewer: Unfortunately, looking at the updated manuscript, I do not see the alterations described above. For example, right in the Summary, lines 20-21, the authors state: "Here we show that the mechanosensitive mammalian homologues of OSCA/TMEM63 channels are monomeric channels gated by lipids...". In Results, lines 77-78: "...we found, regardless of the detergent used for extraction, that hTMEM63A 78 and mTMEM63B are monomeric mechanosensitive channels". As I mentioned before, no experimental data, demonstrating functional relevance of monomeric TMEM63, are presented. Also, gating of TMEM63 by lipids is still a hypothesis, not an established fact.

Response II: The reviewer is indeed correct; we have now completely removed any unambiguous assertion of lipid-gating instead choosing to compare the role

of the lipid-like densities we have identified to lipid-like densities seen in other MS channels. For example, the summary now states:

"Combined our data suggests that the gating mechanism of OSCA/TMEM63 channels may combine structural aspects of the 'lipid-gated' mechanism of MscS and TRAAK channels and the calcium-induced gating mechanism of the TMEM16 family."

We have retained our putative gating model in figure 5.

We have also throughout, including extensively in the discussion, pointed out that while our structures of both TMEM63A and B suggest they are monomers we do not provide unequivocal evidence that is the case in a cellular environment:

"Here we show that the mechanosensitive mammalian homologues of OSCA channels the TMEM63A/B channels are monomeric channels when purified and structurally characterized in a detergent environment, suggesting that one subunit of the OSCA/TMEM63 channel core structure is likely sufficient to form an MS channel."

Comment 4. In Results, the authors state: "...we found, regardless of the detergent used for extraction, that hTMEM63A and mTMEM63B are monomeric mechanosensitive channels..." Again, this does not appear to be correct, because:

a) only LMNG detergent-based protein purification protocol is mentioned in the Methods section. Figure S1b however illustrates FSEC results for TMEM63a with 3 different detergents, but there are no references to it in the Methods. Moreover, Figure S1 is entitled "Purification and cryo-EM data procedure of AtOSCA1.1 in nanodisc", which does not make sense

b) the authors show that mammalian TMEM63 purifies as a monomer and that TMEM63s are monomers in detergent micelles only, not in nanodiscs

c) the manuscript does not contain any functional data on mammalian TMEM63s, e.g. regarding their mechanosensation or oligomeric structure in any environment except detergent micelles.

Generally, the authors often use the term "monomeric hTMEM63A channel" throughout the text, which has not been functionally proven.

Response: We appreciated the reviewer's comment and apology for these mistakes. We added the different detergent extraction protocols in the methods section and revised the Supplementary Fig. 1.

Reviewer: While Figure S1 title has been corrected and the Methods section now contains the appropriate details, the statement about "TMEM63 monomeric mechanosensitive channels" (lines 77-78) remains experimentally unsupported. Figure S1 also contains a number of inconsistencies and lacks important information (see details below).

Response II: We have now modified Figure S1 to remove any inconsistencies and added units to the axis. We also tone down the assertion of monomeric

assembly as above choosing to say that TMEM63A/B “are monomeric channels when purified and structurally characterized in a detergent environment.” We also discuss the fact that hTMEM63A is still a monomer regardless of the detergent used for purification but in the discussion add the following caveat:

“Specifically, we show that two TMEM63 homologues from different species purified with different detergents seem to exist as monomers. Despite this strong evidence and the clear fact that the M10 helix in our TMEM63 structures rotates into the space usually occupied by the OSCA channel central cavity, a fact that would preclude a similar dimeric organization of TMEM63, we of course cannot rule out that in a native lipid or cellular environment that TMEM63 channels organize as dimers with an alternate dimeric interfacial organization. Indeed, on this point it will be interesting to see whether alternate structural approaches including nanodisc reconstitution of TMEM63 homologues reveal similar monomeric organization.”

Comment 5, In Results, the authors state: “...we restrict our below discussion to hTMEM63A...” without specifying the reason. Although the density map of hTMEM63A seems to be better than that of mTMEM63B (Figure S3), the text specifies the opposite and refers to Figure S1, which however contains AtOSCA1.1 data. The authors should fix that, as well as a few similar typos and broken references throughout the text. Related to the previous point - there are two figures entitled almost identically: “Extended Data Fig. 1 | Purification and cryo-EM data processing procedure of AtOSCA1.1 in nanodisc” and “Extended Data Fig. 4 | Cryo-EM data processing procedure of AtOSCA1.1 in nanodisc”. This is very confusing.

Response: Thank you for reminding us of the call to Figure and overlooking in our previous manuscript. We have carefully revised these issues in the revised manuscript accordingly, and have re-scrutinized the whole manuscript. We are sorry for the mistake in the figure legend of Supplementary Fig. 1-2. We have fixed it in the revised manuscript.

Reviewer: While the figures titles and references are now (mostly) fixed, there are still multiple typos and errors in the text. Addressing the first part of my comment - lines 82-83 indicate 4A resolution for hTMEM63A and 3.3A resolution for mTMEM63B, which is the opposite to the data presented in figures S2 and S3, that report higher resolution of hTMEM63A structure. This is probably the reason why the authors focused on a human protein (although it is not mentioned in the text).

Response II: We have now explicitly stated that we focus on hTMEM63A due to the higher resolution structure that we obtained and correctly refer to the relevant supplementary figures. In addition, we have carefully revised the manuscript for typographical and grammatical abnormalities.

Comment 7, The authors further state: “...Structural comparison of the monomeric hTMEM63A and AtOSCA1.1, demonstrates that the M10 of hTMEM63A is rotated into the ‘central cavity’ and would sterically hinder the dimeric interface (Extended Data Fig. 3d-e), providing a likely

reason for the monomeric form.” It is very difficult to understand from the figure why it is the case and which non-conserved residues from the dimer interface could be responsible for that. Again, could not that be an artifact of detergent micelle vs lipid nanodisc systems?

Response: Our previously solved conformation of dimeric AtOSCA1.1 in a detergent environment shows the dimer interface of the AtOSCA1.1 is formed by electrostatic and hydrogen bonding networks (Q and T repeats and R-E pairs) and M10 is isolated by lipids (Zhang etc. NSMB 25, 850–858 (2018)). However, the mammalian OSCA/TMEM63s are not conserved at the electrostatic and hydrogen bonding networks (Q and T repeats and R-E pairs) (Supplementary Fig. 4f). And the structure comparison of dimeric AtOSCA1.1 and monomeric hTMEM63A (Supplementary Fig. 4d-e) shows that there is a rotation of TM10 in hTMEM63A compared to that in AtOSCA1.1, which sterically hinders the formation of the dimer interface. And all structures mentioned above are obtained in a detergent environment. Therefore, it is difficult for mammalian OSCA/TMEM63s to form dimer channels.

Reviewer: Unfortunately it is difficult to agree with the authors. First, the manuscript lacks any discussion or at least a description (similar to the mentioned above) of the interactions maintaining dimeric structure of AtOSCA1.1. Second, although the authors highlight the differences between a dimer interface of AtOSCA1.1 and a corresponding region in hTMEM63A (illustrated in Figure S4), the possibility of forming a different set of interactions between hTMEM63A monomers is not discussed either. In addition to that, Figure S4 itself suffers from a lot of inconsistencies (more details below), which significantly decreases its illustrative value.

Response II: We have extensively revised Figure S4 and most importantly its accompanying legend (see below for all details). Importantly Figure S4 primarily aims to show 4 things;

1. An Architectural comparison of hTMEM63A and mTMEM63B (panels a-c).
2. Clear presentation of the different N-terminus of hTMEM63A that shields the extracellular side of the channel when compared to AtOSCA1.1 (panels d-f).
3. Comparison of some of the residues that coordinate the putative plug lipid and whether they are conserved in OSCA1.1 (panels g-i).
4. The position of the putative plug lipid in TMEM63A with a comparison to the binding site of Ca²⁺ in TMEM16A – to show how similar the location is between these protein families (panels j-k).

We agree M10 conceivably could form a different set of interactions at a putative dimer interface perhaps more like TMC channels. In order to point this out to readers we have now inserted prose around this point within the discussion. However, given we have both TMEM63A and B structures in monomeric forms we prefer not to speculate on what potential dimeric interfaces may exist.

As a final point we of course agree that detergents may influence the oligomeric state of TMEM63A but we think the fact that the monomeric form is seen in 3 different detergent combinations really provides strong support for the idea that TMEM63 channels are monomers. For further elaboration on how we have toned down and discussed the caveats of these claims see comment 2.

Comment 9, Figure 2. The lipid density presented on 2a is very difficult to see (also no σ is reported). On 2f, which is supposed to show the lipid-protein interactions, it is difficult to identify (electrostatically) interacting residues, responsible to lipid molecule docking to this particular location.

Response: We have modified figures 2a and 2f shown below to make the lipid density and lipid binding site clearer.

Reviewer: OK, however there is no information (either in the figure legend or Methods) on what lipid molecule was used for modeling.

Response II: We modelled the lipid using POPC.

Comment 10, An extended state of AtOSCA1.1 in lipid nanodiscs is driven by lyso- lipids. The mechanism of "promotion" of OSCA channel activation, suggested in this paragraph, is not quite clear. The authors suggest that lyso-lipids, bound within the central cavity, facilitate gating. Following this logic, hydrophobic mutation H651A (as shown on Figure 3g) should theoretically increase interaction with a lyso-lipid hydrophobic tail to stabilize the extended conformation and, therefore, to facilitate gating even more. Patch-clamp data (Figure 3h-j), however, clearly indicates the opposite – namely, lower tension sensitivity. Could the authors elaborate more on that?

Response: We thank the reviewer's suggestion. The H651A may impair the putative lyso-lipid and channel interaction thus exhibiting lower tension sensitivity.

Reviewer: The authors call H651 residue a "putative lyso-lipid coordinating residue" (line 141), however, Figure 3G hardly illustrates that statement. Neither interacting (electrostatically in case of ionized histidine side chain or hydrophobic if it is not) residues are labeled, nor the distance between any atoms are shown.

Response II: We agree with the reviewer that the panel as previously presented was both unclear and unconvincing. We have now added a modified version showing the putative hydrophobic interaction of an unprotonated His651 with the lipid acyl chain. Distances are added for further information and other potential residues that may coordinate this lipid are also shown for completeness. The point raised by the reviewer also leads to the interesting question of how protonation of His651 may influence mechanosensitivity and we are working on this very point now but hope the reviewer agrees this is beyond the scope of this current manuscript.

Comment 11, Figure 4 does not seem to illustrate the results and conclusions presented in the text. Figures 4a and 4b represent contracted and extended states of OSCA3.1, while Figures 4c and 4d are superposition of 4a and 4b viewed from different angles. Residues labeled in red probably refer to the central cavity cross-section illustrated on Figure 4F, but there are no references to them in the legend, as there is no legend for Figure 4e. The authors state, that single monomers from the both (contracted and expanded) states

superimpose and are basically identical, but there is no figure showing that. The interaction between L487 (or L478 as mentioned in the legend?) residues from TM5-TM6 linkers (shown on Figure 4e) is not elaborated in the legend. In particular, it is not clear which state is illustrated, as no distance between these residues is labeled. No differences between leucines interaction in contracted and extended states are illustrated either.

Response: Thanks for this suggestion. We have modified Figure 4 and its legend as the reviewer's suggestion.

Reviewer: Unfortunately, though the modifications of Figure 4 addressed most of concerns, there are still several issues with this figure (more details below).

See comments and amendments as below.

Comment 12, Maybe I am missing something, but the presentation of data in the manuscript seems to me quite inconsistent. For example, though the authors discuss hypothetical regulation of OSCA1.1 by lyso-lipids, they do not consider OSCA3.1 or TMEM63. In fact, it is unclear from the text even if any lipid-like densities were detected in OSCA3.1/TMEM63 in the locations, where those were found for OSCA1.1. Vice versa, speaking about hypothetical "gating" of TMEM63 by lipids, the authors completely omit any discussion concerning structurally related OSCA1.1 and OSCA3.1. Unfortunately, such an approach turns the manuscript into a set of poorly related parts.

Response: We appreciated the reviewer's comments. In the corresponding region of hTMEM63a, there is a similar plug-lipid in OSCA1.1 and OSCA3.1 albeit the density is not better than hTMEM63a. We add the potential plug-lipid density in OSCA1.1 and OSCA3.1 in the supplementary figure 8. And we discussed the structurally related hTMEM63a, OSCA1.1 and OSCA3.1 in the first discussion part.

Reviewer: It is not quite clear what the authors mean in lines 176-179. Are the lipid-like densities found in AtOSCA structures weaker than that from hTMEM63A? If so, how significant is the difference? Do authors suggest that the stronger densities imply stronger protein-lipid interactions, which result in decreased tension sensitivity? Hypothesized lipid densities presented in Figure 8 are not color-coded and blend into the protein density, so it is very difficult to compare them.

Response II: First, we have now highlighted the lipid-like densities in all our structures to show the potential for the lipid plug to be a universal feature of OSCA/TMEM63 channels (see revised Supp Fig 8). These densities are clearly present in all structures, but they lack the resolution of the TMEM63A lipid-like density. It is of course possible that the reduced mechanical sensitivity of TMEM63A makes it easier to identify this strongly bound putative lipid plug and we have speculated as much in our discussion. We now use this lipid-like density as a means to progress through the manuscript hopefully providing more cohesion between the presented data.

Miscellaneous:

Lines 100-102: F555, F556, and Y559 all belong to M6 helix, but what are the other pore-lining residues (from other helices)?

Response II: In order to make this clearer we have now colour coded residues from different TM helices in Figure 2d-e. We believe this makes it much clearer as to which residues from each TM are contributing to the contouring of the pore.

Line 125: authors mention AtOSCA1.1 structure obtained in detergent, but do not mention any references or PDB.

Response II: We have now noted the reference and the relevant PDB in both the text and on the figure. Specifically, we write:

"The AtOSCA1.1 channel in a lipid nanodisc is in an overall extended state when compared to the detergent environment (PDB:6JPF; Fig. 3a-e)."

Lines 145-150: was AtOSCA3.1 structure obtained also in nanodiscs?

Response II: We now make this unambiguous:

"Our high-resolution cryo-EM data obtained in a detergent environment contain two conformations of AtOSCA3.1 channels (Supplementary Fig. 6a-g, Supplementary Fig. 7a-b)."

Lines 180-181: I do not understand the logic of the statement about a putative (!) modulating lyso-lipid playing however an important role in the activation of dimeric OSCA/TMEM63 channels. If it is not quite clear yet, if there is a modulating lipid, how one can state that it plays an important role?

Response II: Text around this point has been amended.

After the authors fixed the figure legends references and numbers, there are still multiple errors and inconsistencies in figures and figure legends:

Figure 1. It is not clear, which (extended or contracted, nanodiscs or detergent) conformation of AtOSCA1.1 is compared to hTMEM63A. Would be a good idea to include structures' PDB codes too.

Response II: We have now added the relevant codes to make it easier for reviewers to interpret and navigate the data.

Figure 2. Line 371 – "A lipid that plugs..." - how confident the authors are it could not be a detergent molecule?

Response II: The density in hTMEM63A is shown in Figure2A-B and clearly maps well onto our putative POPC. From the densities in the Cryo-EM map it seems much more likely this is a lipid and not a detergent.

Line 374 – how was the pore profile calculated – HOLE, anything else?

Response II: HOLE alone was used to calculate the pore profile.

Line 377 (e) – it is not quite clear why the authors only presented M6 side chains, what about other pore-lining helices (M4, M5)?

Response II: As mentioned above we have now colour coded residues from different TM helices in Figure 2D-E. We believe this makes it much clearer as to which residues from each TM are contributing to the contouring of the pore.

Line 380 – “The lipid is shown...” - modeled with what kind of lipid (DOPC, etc)? The distances in (f) are Å I suppose.

Response II: The plug lipid is modelled as POPC and this has been added to the methods and figure legend. Units of distances have also been added and are shown in Å.

Figure 3. Line 387 – “a-b” should be “a, d” based on what is presented in the figure.

Response II: Correct, we have modified the panel labelling.

Line 394 – “black arrows”.

Response II: Corrected.

Line 396 – the distances are in Å? In (g) – could the authors also provide electron densities for the interacting residues/atoms?

Response II: We have modified Figure 3G.

Figure 4. The last sentence (color coding description) should go to the top. Are the structures obtained in nanodiscs (as AtOSCA1.1)?

Response II: Description is amended as suggested and we now make the point regarding how the structure was obtained unambiguous (as below) and add the relevant information in the legend of Figure 4:

“Interestingly, our high-resolution cryo-EM data obtained in a detergent environment contain two conformations of AtOSCA3.1 channels (Supplementary Fig. 6a-g, Supplementary Fig. 7a-b).”

Line 407 – “black arrows”.

Response II: Corrected

Lines 411-412 – “L478” is mentioned in the legend, but “L487” is labeled in the figure.

Response II: This should be L487 and has been corrected.

Figure 5. Lines 418-419 - “...or dimeric-coupled OSCA/TMEM63 (a)...”, while (a) represents only one monomer.

Response II: Amended to clearly distinguish monomeric TMEM63 from dimeric OSCA.

Figure S1. The graphs lack axis scales (elution volume and fluorescence intensity units). (a) – what detergent was used? Based on the Results text it is probably LMNG? (b) is very confusing: first, black color-coded profile is not annotated (is it DDM, which is supposed to be brown?); second, LMNG profiles for hTMEM63A in (a, brown) and (b, cyan) look different. Would be a good idea to use the same elution volume scale in both (a) and (b) and to mention the column type used.

Response II: Modified and corrected as suggested.

Line 429 – “traces”.

Response II: Corrected.

Line 431 – “...peak are denoted...”.

Response II: Corrected.

Figure S2. Line 442 – “is based”.

Response II: Corrected.

Figure S4. Line 460 – “...superposition of them.”

Response II: Corrected.

(e) – no description of the black dashed arrow, “TM10” in the figure, but “M10” in the legend.

Response II: This was missing and should denote the extended N-terminus and not M10 (see next point).

(e, f) – do the red boxes represent the same region of M10? If this is a full transmembrane helix, its length in the alignment seems to be quite short – only 10 amino acids. Alignment itself also looks very strange, for example, the conserved cysteine (lower panel, second aa from the left, AtOSCA1.1 residue 342) is not labeled as conserved; several letters (M, D, and W) are clipped (also applies to panel (I)), etc.

Response II: We apologise these two panels were completely mislabelled and actually represent the N-terminus of hTMEM63A. Here we are trying to point out that this N-terminal cap region is very different in hTMEM63A compared to the AtOSCA1.1 structure. We have relabelled both panels correctly and now believe this figure is much easier to interpret.

(i) – “N-terminus”.

Response II: Corrected.

(g, h) – what are the surfaces presented? What do their colors (blue and brown) mean? (j, k) – is (j) supposed to be AtOSCA1.1 structure (as AtOSCA1.1 residues are mapped onto it)? Also, color coding is inconsistent with (d, e). If it is indeed mTMEM63B (as the legend states), what is the reason for placing it there, as it is discussed neither in the text nor in the figure legend? (l) – erroneously duplicates (i); no description of color coding of the structures. (n) is described as “mTMEM63B” in the legend; there are no comments on the red balls, probably representing bound Ca²⁺ ions; color coding is inconsistent between (m) and (n); the structures are shown from different angles and are also differently scaled, which makes them nearly impossible to compare.

Response II: The new extensively edited Figure S4 with new legend fully addresses all of these points.

Figure S5. (a) - protein gel ladder is not size-labeled.

Response II: Amended.

Line 477 – “black arrows” as far as I can see.

Response II: Corrected.

Figure S6. Line 496 – “...is based on...”.

Response II: Corrected.

A legend for panel (g) is missing.

Response II: Corrected.

Figure S7. Line 509 – RMSD (not RSMD) in Å I guess is meant here.

What is the color coding in (c)?

Response II: Corrected and colour coding has been added to the figure legend.

Figure S8. I think it would be a very good idea to use a different color for the hypothesized lipid densities in all panels, otherwise they blend with a protein density and are almost impossible to see.

Response II: This is an exceptionally pertinent point. Here we have coloured the lipid-like density in red. We have tried to make it as obvious as possible. We note of course that each of these densities can easily be seen in our EMDB deposited maps especially the putative lipid in the *h*TMEM63A structure.